# SCENEWEAVER: All-in-One 3D Scene Synthesis with an Extensible and Self-Reflective Agent

**Yandan Yang**[1,*]  **Baoxiong Jia**[1,*,†,✉]  **Shujie Zhang**[1,2]  **Siyuan Huang**[1,✉]

[1] State Key Laboratory of General Artificial Intelligence, BIGAI    [2]Tsinghua University

https://scene-weaver.github.io/

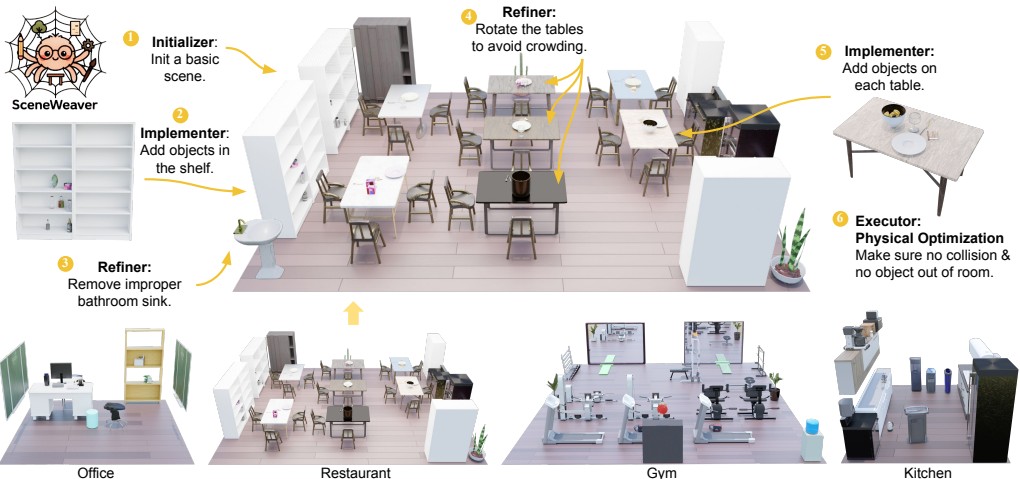

Figure 1: **Overview of SCENEWEAVER**, a reflective agentic framework built on standardized and extensible tool interfaces that unifies the strengths of existing scene synthesis methods to produce visually realistic, physically plausible, instruction-aligned 3D scenes.

## Abstract

Indoor scene synthesis has become increasingly important with the rise of Embodied AI, which requires 3D environments that are not only visually realistic but also physically plausible and functionally diverse. While recent approaches have advanced visual fidelity, they often remain constrained to fixed scene categories, lack sufficient object-level detail and physical consistency, and struggle to align with complex user instructions. In this work, we present SCENEWEAVER, a reflective agentic framework that unifies diverse scene synthesis paradigms through tool-based iterative refinement. At its core, SCENEWEAVER employs a language model-based planner to select from a suite of extensible scene generation tools, ranging from data-driven generative models to visual- and LLM-based methods, guided by self-evaluation of physical plausibility, visual realism, and semantic alignment with user input. This closed-loop reason-act-reflect design enables the agent to identify semantic inconsistencies, invoke targeted tools, and update the environment over successive iterations. Extensive experiments on both common and open-vocabulary room types demonstrate that SCENEWEAVER not only outperforms prior methods on physical, visual, and semantic metrics, but also generalizes effectively to complex scenes with diverse instructions, marking a step toward general-purpose 3D environment generation.

---

[*] Equal contribution. [†]Project lead. [✉] Corresponding authors.

39th Conference on Neural Information Processing Systems (NeurIPS 2025).

# 1 Introduction

3D scene synthesis [1, 2, 3, 4, 5, 6, 7, 8, 9, 10, 11] has been a long-standing research topic in computer vision and graphics, primarily focused on generating visually realistic 3D environments for applications such as interior design, virtual content creation, and gaming asset creation. With the recent rise of embodied artificial intelligence (EAI), the scope of scene synthesis has naturally expanded to accommodate new functional demands [12, 13, 3]. Beyond achieving **visual realism**, scenes are now expected to be **physically interactable** within simulators and **precisely controllable** in response to task-specific user instructions, particularly in constructing tailored environments for training and evaluating embodied agents. These extended requirements pose significant new challenges for 3D scene synthesis.

Despite rapid progress, existing methods fall short of holistically addressing the requirements for realistic, controllable, and physically plausible scene synthesis, as summarized in Tab. 1. Rule-based systems [12, 14] ensure physical validity through hand-crafted constraints, but lack extensibility across diverse scene types and offer limited controllability due to their rigid, manually defined logic. Data-driven generative learning methods [15, 2, 3], while more flexible, are constrained by the scarcity of high-quality, scene-level 3D datasets (*e.g.*, 3D-Front [16]). As a result, they typically produce visually realistic scenes within pre-defined categories but generalize poorly to novel scene types or layout instructions. Methods based on Large Language Models (LLMs) approaches [7, 8, 9, 10, 11] offer stronger open-vocabulary understanding and semantic flexibility, yet often struggle with spatial reasoning and 3D awareness, resulting in physically implausible rearrangements. Collectively, these limitations highlight that *no single approach is sufficient to meet the combined demands of realism, physical plausibility, and controllability*. This motivates the need for a comprehensive and adaptable scene synthesis framework capable of synthesizing high-quality 3D scenes.

Inspired by recent advances in LLM-based agents, which demonstrate strong reasoning and planning capabilities in complex tasks, recent works in 3D scene synthesis have begun to move beyond monolithic approaches by decomposing the generation process into sequential compositions of modular synthesis components, forming multi-step pipelines coordinated by LLMs. A common strategy starts with generating coarse, scene-level layouts through interaction with LLMs [10, 8, 9, 7], followed by progressive refinement using pre-trained 2D generative models or Multi-modal LLMs (MLLMs) for asset generation [17, 18], object placement [19, 20, 21], and texture inpainting [11, 22]. While these pipelines leverage both the specialization of individual models and the semantic flexibility of MLLMs, they remain largely "static", *i.e.*, their planning and execution are governed by fixed prompts and hard-coded module invocation logic over a limited set of synthesis tools. This design overlooks the potential to couple reasoning with adaptive decision-making based on generation feedback, and the ability to seamlessly integrate diverse synthesis tools through a unified interface. As a result, these systems fall short of enabling self-refining and extensible agents, leaving the full potential of multi-modal foundation models underutilized.

To address the aforementioned challenges, we propose SCENEWEAVER, a reflective agentic framework that enables MLLMs to synthesize 3D scenes in a self-refining manner through a set of easily extensible tool interfaces. Specifically, SCENEWEAVER consists of two core components: 1) a standardized and extensible tool interface that abstracts diverse scene synthesis methods into modular tools operating at different levels of generation granularity; 2) a self-reflective planner that dynamically selects tools and iteratively refines the scene by reasoning over feedback from previous generations, while applying the planned modifications and enforcing physical plausibility with a physics-aware executor. This framework enables closed-loop, feedback-driven scene evolution, where the agent identifies areas for improvement, invokes appropriate tools, and updates the scene under physical constraints. Extensive experiments show that SCENEWEAVER achieves new state-of-the-art across a broad range of scene types and open-vocabulary instructions, demonstrating strong visual realism, physical plausibility, and precision in instruction following. We also provide ablation studies showing that the self-refining design is critical to achieving high-quality scene synthesis and that integrating diverse tools leads to significant performance improvement compared to monolithic approaches. In summary, our contributions are as follows:

- We propose SCENEWEAVER, the first reflective agentic framework for 3D scene synthesis, enabling MLLMs to iteratively refine scenes through feedback-driven planning with modular tools.

Table 1: **Comparison of different scene synthesis methods.** A single approach is not sufficient to meet the combined demands of realism, physical plausibility, and controllability, which motivates the need for a comprehensive and adaptable scene synthesis framework.

| Previous Work | Physical Plaus. | Small Object | Open Vocab. | #Room Type | Real | Accurate | Large Scale | CAD Source | Developing Platform | Method |
|---|---|---|---|---|---|---|---|---|---|---|
| ATISS [15] | ✗ | | | | | | | | - | |
| DiffuScene [2] | ✗ | ✗ | ✗ | 3 | ✓ | ✗ | ✓ | 3D FUTURE | - | Model-based |
| PhyScene [3] | ✓ | | | | | | | | - | |
| Infinigen [14] | ✓ | ✓ | ✗ | 5+ | ✗ | ✓ | ✓ | Generated | Blender | Rule-based |
| Procthor [12] | ✓ | ✓ | | 4 | | | | RoboTHOR | AI2-THOR | |
| MetaScene [19] | ✓ | ✓ | ✓ | 30+ | ✓ | ✗ | ✗ | Mixed | - | Vision-based |
| ACDC [20] | ✓ | ✓ | ✓ | Unlimited | ✓ | ✗ | ✓ | Behavior | OmniGibson | |
| Architect [17] | ✓ | ✓ | ✓ | Unlimited | ✓ | ✗ | ✓ | Mixed | - | |
| LayoutGPT [9] | ✗ | ✗ | | | | | | 3D FUTURE | - | |
| Holodeck [10] | ✓ | ✓ | | | | | | Mixed | AI2-THOR | |
| AnyHome [11] | ✗ | ✗ | ✓ | Unlimited | ✓ | ✗ | ✓ | Generated | - | LLM-based |
| I-Design [7] | ✓ | ✗ | | | | | | Objaverse | - | |
| LayoutVLM [8] | ✓ | ✗ | | | | | | Objaverse | - | |
| **SCENEWEAVER** | ✓ | ✓ | ✓ | Unlimited | ✓ | ✓ | ✓ | Mixed | Blender / IsaacSim | Unified |

- SCENEWEAVER introduces a comprehensive reason-act-reflect paradigm that formalizes the planner's decision making, reflection, and action protocols, along with a standardized and extensible tool interface for synergizing diverse scene synthesis methods based on their respective strengths.
- Extensive experiments on open-vocabulary scene synthesis demonstrate that SCENEWEAVER outperforms existing methods in both visual realism, physical plausibility, and instruction following. We also provide meticulously designed ablation studies to highlight the effectiveness of the proposed reflective agentic framework.

## 2  Related work

**3D Indoor Scene Synthesis**  3D indoor scene synthesis is typically formulated as a layout prediction task, where objects are represented by 3D bounding boxes and semantic labels [16, 15, 8]. Data-driven generative models [15, 2, 3], trained on datasets like 3D-FRONT [16], learns realistic but coarse scene layouts, constrained by the limited variety and level of detail of scenes in the dataset. To address this limitation, recent work leverages language and 2D foundation models to provide missing priors on scene types and fine-grained details. LLM-based methods [7, 8, 9, 10, 11] combine textual prompts with rule-based systems [12, 14] to generate diverse scenes, but often suffer from hallucinations and the poor spatial reasoning capability of LLMs. Meanwhile, methods based on 2D foundation models improve scene detail and spatial coherence through image-conditioned generation [17, 18] or real-to-sim conversions [20, 19]. However, they remain limited by the capability of image generation models and challenges in 2D-to-3D lifting, exhibiting semantic or physical inconsistencies under complex scene generation instructions. Overall, no existing paradigm sufficiently balances realism, physical plausibility, and controllability. To this end, we propose SCENEWEAVER, a unified and extensible self-reflective agentic framework that integrates the complementary strengths of existing approaches for high-quality 3D scene synthesis.

**Spatial Reasoning of MLLMs**  Recent works have explored using the reasoning and generative abilities of MLLMs for 3D scene synthesis. To address their limitations in spatial reasoning, these methods often incorporate structured constraints and external logic to enhance physical plausibility. Some approaches apply rule-based constraints as post-processing to correct implausible object placements [10], while others adopt multi-agent or role-based decomposition to reduce hallucinations and improve coherence [7]. Additionally, efforts have been made to guide generation through scene-aware tools, such as programmatic layout representations [8] or geometric reasoning modules [23]. Although these systems MLLMs with reasoning chains and post-optimization mechanisms, they typically rely on fixed toolsets and predefined constraints, limiting their flexibility and extensibility. In contrast, SCENEWEAVER isdesigned to support a diverse and extensible set of tools through a standardized interface, enabling dynamic tool selection and composition via a reflective planning mechanism for reasoning-driven 3D scene synthesis.

**LLM-based Agentic Framework**  A growing body of work leverages LLMs as autonomous agents for complex tasks across domains such as scientific discovery [24], clinical decision-making [25], and visual reasoning [26]. As LLMs' reasoning capabilities gradually advance, the focus has shifted from narrow task-specific agents to general-purpose agentic frameworks [27, 28, 29, 30] that

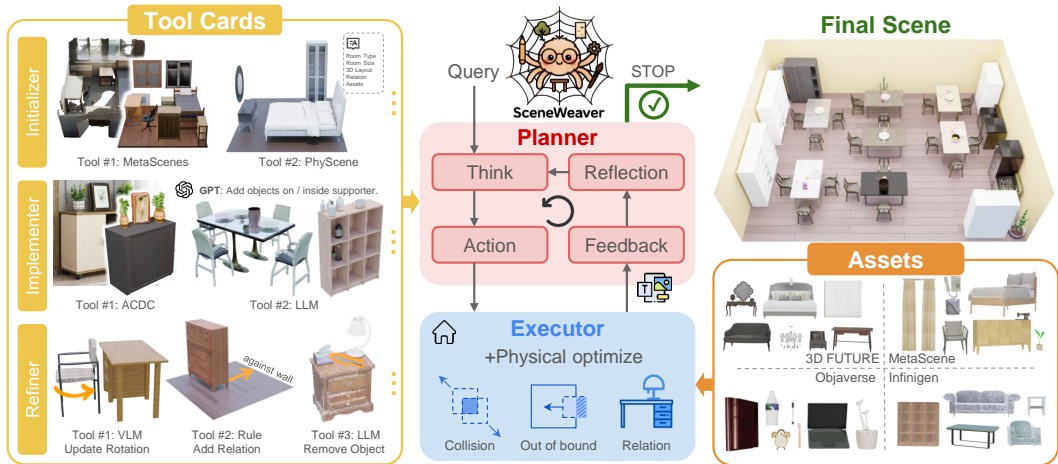

Figure 2: **The SCENEWEAVER pipeline.** Following a reason–act–reflect paradigm, SCENEWEAVER iteratively refines scenes by integrating the strengths of diverse scene synthesis tools.

coordinate multiple specialized tools to solve complex problems collaboratively. Recent work [31] has demonstrated that the extensibility and planning capabilities of LLMs, *i.e.*, the ability to flexibly integrate diverse tools and coordinate them effectively, are crucial for solving complex reasoning tasks and lead to significant performance gains. However, these insights on agent development remain largely underexplored in the context of 3D tasks. Motivated by its relevance to 3D scene synthesis, SCENEWEAVER draws on advances in LLM-based agentic frameworks and adopts the OpenManus [30] platform to implement an agentic framework, with a particular emphasis on extensibility of tools and the reason-act-react paradigm for 3D scene synthesis.

## 3   The SCENEWEAVER Framework

In this section, we present the design of SCENEWEAVER, an agentic framework that enables LLMs to perform feedback-guided, self-reflective 3D scene synthesis using a diverse set of scene synthesis tools. The SCENEWEAVER framework comprises two key components: 1) a **standardized tool interface** that organizes the majority of existing scene synthesis methods into modular tools categorized by their synthesis granularity (Sec. 3.1); 2) a **self-reflective planner** that dynamically selects tools, iteratively refines the scene based on feedback, and performs physics-based optimization to enhance physical plausibility. An overview of SCENEWEAVER is provided in Fig. 2.

Before describing each component, we formalize the overall problem setup. Given a user query $q \in \mathcal{Q}$ and a tool set $\mathcal{D} = \{d_i\}_{i=1}^n$, SCENEWEAVER aims to synthesize a 3D scene $s_T$ through $T$ iterative refinement steps. Each scene state $s_t$ is represented by both 3D layout information and also 2D renderings from selected camera views (as illustrated in Sec. A.1). At each step $t \in [1, ..., T]$, the self-reflective planner receives a reflection $v_{t-1}$ including quantitative scores and explanatory justifications assessing the quality and instruction alignment of the previous scene $s_{t-1}$. Based on this feedback, the planner selects a tool $d_t \in \mathcal{D}$ to refine, and the physics-aware executor applies the refinement and performs physical optimization to produce the updated scene $s_t$. A new reflection $v_t$ is then computed for $s_t$, and the process repeats.

### 3.1   Standardized Scene Synthesis Tool Interface

**Tool Catalog**   As summarized in Tab. 1, existing 3D scene synthesis methods vary widely in their design and focus. To leverage their complementary strengths within a unified framework, we introduce a standardized tool interface that abstracts each method as a modular synthesis tool. These tools are categorized according to their synthesis granularity:

- *Scene Initializer:* This class of tools generates full-scene layouts and serves as the starting point for synthesis. We categorize initializers into three types: 1) data-driven generative models [15, 2, 3], which offer scalable generation learned from human-designed indoor scene datasets but are limited to pre-defined scene types; 2) real-to-sim methods [20, 19] that create digital twins or cousins

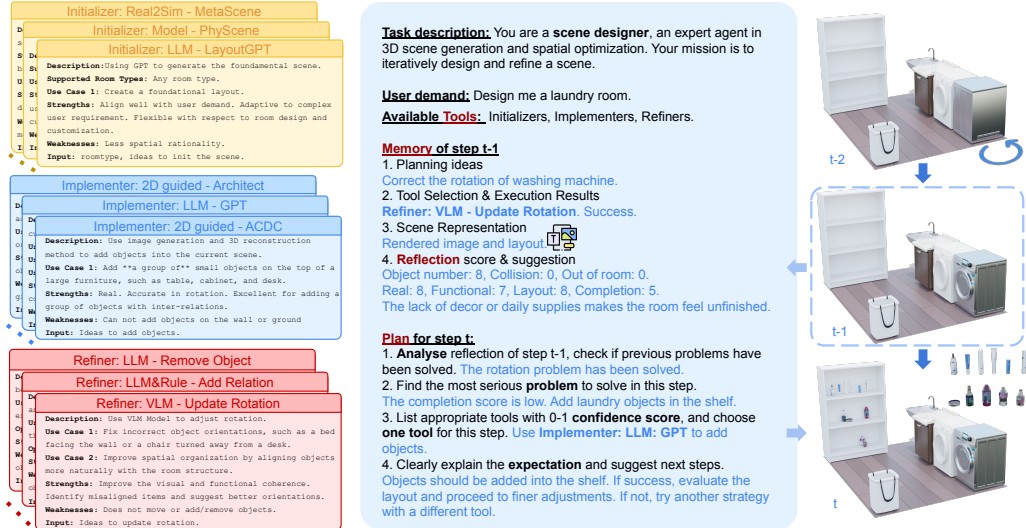

Figure 3: **A visualization of standardized tool interfaces and the reflective planning process.** The self-reflective planner leverages diverse tools to first correct the misoriented laundry machine and then enhance scene details by adding small objects to the shelf (right).

of realistic scenes, providing detailed high-quality scenes but with limited diversity and scale; 3) LLM-based [9, 10, 8, 7] that enable open-vocabulary and flexible generation from natural language, but often exhibit semantic or physical inconsistencies due to limited spatial reasoning.

- *Microscene Implementer:* This class of tools adds micro scene details (*e.g.*, small objects placed on desks or shelves) that are often missing from whole-scene synthesis methods. We consider two types of implementers: 1) LLM-based tools that generate microscene layouts conditioned on local context (*e.g.*, placing a keyboard and monitor on a desk), offering semantic diversity but prone to spatial placement errors (*e.g.*, misaligned or backward-facing objects); and 2) 2D-guided tools [17, 20], which synthesize reference images of microscene regions using pre-trained 2D generators, then mapping corresponding 3D assets to the predicted layout. While still constrained by the spatial reasoning capability of 2D models, the 2D-guided tools enhances visual realism and relative spatial coherence between objects.

- *Detail Refiner:* While previous tools synthesize scenes at various granularities, they often introduce errors such as object misplacement or implausible configurations. Refiner tools address these issues by enforcing constraints and refining object poses. First, we extend on rule-based scene synthesis methods [12, 14] and use LLMs to convert user queries into relational constraints that guide object placement. Second, to compensate for layout generators that neglect object orientation and scale, we incorporate dedicated tools to refine objects' full 6D pose (location, rotation, scale). Finally, an LLM-based remover identifies and eliminates semantically incorrect or severely misplaced objects.

**Standardized Tool Cards**   To ensure flexible integration of new tools into SCENEWEAVER, we define standardized tool cards that guide the planner in deciding when and how to invoke each synthesis method based on their specialized strengths. Examples are shown in Fig. 3. Each tool card contains mandatory fields, including tool description, applicable scenarios, usage constraints, and required input parameters. We also include example usage and tool-specific strengths to help the planner select the most appropriate tool based on user queries or iterative feedback. For initializer tools, supported room types are listed to reflect model-specific limitations and enable the agent to infer room types from queries when evaluating tool applicability. This modular design ensures seamless integration, extension, and replacement of scene synthesis methods without modifying the overall agentic framework.

## 3.2   Feedback-driven Self-reflective Planning

**Reflection Generation**   To support self-reflective planning in SCENEWEAVER, we first define the process for generating self-evaluated feedback over synthesized scenes. Specifically, given a generated scene $s_t$, we invoke an MLLM (*e.g.*, GPT-4) to produce a reflection $v_t$ comprising two components: 1) physical metrics, including collision scores, room boundary violations, and object count and diversity; and 2) perceptual metrics, including visual realism, functionality, layout

coherence, alignment with the user query, and scene completeness. In addition to scalar scores, the MLLM generates natural language justifications and improvement suggestions as input to the planner. This feedback forms a core reasoning signal for the planner in the subsequent step, enabling it to assess tool effectiveness and adapt its strategy accordingly. If the feedback indicates abnormal degradation (*e.g.*, sharp drops in quality or constraint violations), the planner can roll back and replan the current step.

**Self-reflective Planning**   Given the user query $q$, a tool set $\mathcal{D}$, and memory of previously selected tools, generated scenes, and reflection feedback $m_t = (d_{t-l:t-1}, s_{t-l:t-1}, v_{t-l:t-1})$, where $l$ determines the length of memory, the planner in SCENEWEAVER determines the most appropriate refinement action. Leveraging context-aware function-calling capabilities in LLMs, the planner first summarizes the current context (*i.e.*, memory) and identifies the most critical problem to address at step $t$. It then ranks candidate tools by suitability and confidence, selects the most promising tool $d_t \in \mathcal{D}$, and generates tool-specific instructions (*e.g.*, "populate empty tables with contextually relevant objects"). Tool confidence scores are dynamically adjusted based on past performance, *i.e.*, failures reduces confidence, and repeated failures trigger replanning by reprioritizing refinement targets and selecting alternative tools. Our self-reflective planner is built on the OpenManus platform [30], following the ReAct-style [32] reasoning and planning pipeline. We provide illustrative examples in Fig. 3 and more detailed prompts in Sec. A.2.

**Physics-aware Execution of Plans**   Since most tools described in Sec. 3.1 operate on 3D bounding box layouts, a physics-aware executor is required to replace these drafts with concrete 3D assets and enable both physical post-optimization and accurate evaluation of physical metrics. To this end, we build our executor on top of Infinigen [14] and Blender. At each iteration $t$, the executor loads the previous scene and the layout modifications proposed by tool $d_t$, then retrieves and replaces object instances with 3D meshes from a collected asset pool combining Objaverse [33], 3D-Future [34], Inifinigen [14], *etc.*, depending on the tool. To ensure spatial consistency with relationships or constraints generated by *detail refiner* tools, the executor also adjusts object placements to satisfy relational constraints (*e.g.*, aligning chairs to face desks) produced by the detail refiner. It then performs a fixed number of physics-based optimization steps to resolve collisions and boundary violations. We provide additional implementation details in Sec. A.5.

# 4   Experiment

In our experiments, we aim to answer the following questions: **Q1:** How does SCENEWEAVER perform compared to existing data-driven and open-vocabulary scene synthesis methods? **Q2:** How does the reflective agentic framework behave during the iterative scene refinement? **Q3:** How effective is each module in SCENEWEAVER, and how critical are they to overall performance?

**Settings**   We quantitatively evaluate SCENEWEAVER against existing methods under two primary settings: 1) common room types, where large-scale human-designed datasets support direct data-driven learning, and 2) open-vocabulary scene generation, following [8], which evaluates generation across diverse room type descriptions. In the common setting, models are evaluated based on the average score over 10 scenes each for the living room and bedroom categories. In the open-vocabulary setting, evaluation is based on the average score over 3 scenes for each of 8 room types, using the prompt "Design me a `<room_type>`" as the user query. We also include a setting with complex queries to assess SCENEWEAVER's fine-grained controllability over scene generation, with details provided in Sec. 4.3. For all settings, we set the maximum number of iterations in SCENEWEAVER to 10. The memory length is set to 1 to avoid hallucination. We provide additional experimental details in Sec. B.

**Baselines**   For the common settings, we compare with data-driven scene synthesis models including ATISS [15], DiffuScene [2], and PhyScene [3]. For these three methods, we train the model over the 3D-Front [16] dataset following the conventional learning evaluation schemes. We also compare with state-of-the-art open-vocabulary 3D scene synthesis methods including LayoutGPT [9], Holodeck [10], and I-Design [7] on both the common and the open-vocabulary setting. As LayoutGPT was originally limited to bedrooms and living rooms, we adapt it to open-vocabulary room types by modifying its prompts and constraints. To evaluate the final scene quality, we retrieve assets from Objaverse [33] using OpenShape [35] text embeddings following [7].

Table 2: **Quantitative comparison on common room types** between SCENEWEAVER and existing scene synthesis methods. For LLM-based methods, we use "Design me a `<room_type>`" as the user query.

| Method | Bedroom | | | | | | | Living Room | | | | | | |
|---|---|---|---|---|---|---|---|---|---|---|---|---|---|---|
| | Physcis | | | Visual & Semantics | | | | Physcis | | | Visual & Semantics | | | |
| | #Obj ↑ | #OB ↓ | #CN ↓ | Real. ↑ | Func. ↑ | Lay.↑ | Comp. ↑ | #Obj ↑ | #OB ↓ | #CN ↓ | Real. ↑ | Funtc. ↑ | Lay.↑ | Comp. ↑ |
| ATISS [15] | 3.9 | 0.5 | 0.6 | 7.4 | 7.1 | 6.6 | 4.2 | 7.8 | 0.1 | 0.7 | 5.8 | 5.3 | 6.4 | 3.7 |
| DiffuScene [2] | 3.5 | 0.1 | 1.1 | 6.5 | 7.0 | 6.7 | 3.6 | 6.9 | 0.5 | 1.2 | 5.5 | 4.9 | 5.2 | 3.5 |
| PhyScene [3] | 3.3 | 0.1 | 0.3 | 5.7 | 6.3 | 5.7 | 4.0 | 8.0 | 0.0 | 0.7 | 5.2 | 5.3 | 5.1 | 3.3 |
| LayoutGPT [9] | 5.4 | 1.0 | 1.3 | 7.5 | 8.1 | 6.7 | 4.2 | 8.4 | 1.1 | 2.8 | 6.4 | 5.8 | 5.2 | 3.6 |
| Holodeck [10] | **32.2** | 0.0 | 38.5 | 8.6 | 9.1 | 7.8 | 6.2 | **23.0** | 0.0 | 5.3 | 8.9 | 9.3 | 7.6 | 8.1 |
| I-Design [7] | 9.6 | 0.0 | 0.0 | 8.6 | 9.3 | 7.6 | 6.1 | 9.7 | 0.0 | 0.0 | 8.4 | 8.9 | 7.7 | 5.9 |
| Ours | 14.0 | **0.0** | **0.0** | **9.2** | **9.8** | **8.4** | **9.4** | 17.3 | **0.0** | **0.0** | **9.1** | **9.5** | **8.0** | **8.7** |

Table 3: **Quantitative comparison on open-vocabulary generation** between SCENEWEAVER and existing methods. We report the average score across 8 scene types to evaluate overall model performance.

| Method | Bathroom | | | | | | | Children Room | | | | | | | Gym | | | | | | |
|---|---|---|---|---|---|---|---|---|---|---|---|---|---|---|---|---|---|---|---|---|---|
| | #Obj | #OB | #CN | Real. | Func. | Lay. | Comp. | #Obj | #OB | #CN | Real. | Func. | Lay. | Comp. | #Obj | #OB | #CN | Real. | Func. | Lay. | Comp. |
| LayoutGPT | 7.7 | 1.3 | 1.0 | 8.3 | 9.3 | 7.7 | 6.0 | 7.3 | 1.0 | 0.7 | 6.3 | 8.0 | 6.0 | 4.0 | 6.7 | 0.7 | 0.0 | 6.7 | 6.7 | 5.7 | 3.7 |
| Holodeck | 12.0 | 0.0 | 1.7 | 7.7 | 6.7 | 7.0 | 5.3 | 13.7 | 0.0 | 2.0 | 7.5 | 7.5 | 6.5 | 5.5 | 20.3 | 0.0 | 5.3 | 9.7 | 9.3 | 6.7 | 6.0 |
| I-Design | 9.7 | 0.0 | 0.0 | 7.4 | 7.2 | 7.4 | 5.4 | 11.3 | 0.0 | 0.0 | 7.8 | 8.3 | 6.8 | 5.5 | 12.0 | 0.0 | 0.8 | 8.2 | 8.4 | 7.0 | 5.2 |
| Ours | 19.7 | 0.0 | 0.0 | 9.0 | 10.0 | 8.0 | 9.0 | 23.0 | 0.0 | 0.0 | 9.0 | 10.0 | 8.3 | 8.3 | 29.7 | 0.0 | 0.0 | 9.0 | 10.0 | 8.0 | 7.3 |

| Method | Meeting Room | | | | | | | Office | | | | | | | Restaurant | | | | | | |
|---|---|---|---|---|---|---|---|---|---|---|---|---|---|---|---|---|---|---|---|---|---|
| | #Obj | #OB | #CN | Real. | Func. | Lay. | Comp. | #Obj | #OB | #CN | Real. | Func. | Lay. | Comp. | #Obj | #OB | #CN | Real. | Func. | Lay. | Comp. |
| LayoutGPT | 7.3 | 1.0 | 0.7 | 4.0 | 3.0 | 5.3 | 2.0 | 7.3 | 0.3 | 0.0 | 6.7 | 7.7 | 6.3 | 4.0 | 7.0 | 0.3 | 1.7 | 3.3 | 2.3 | 4.7 | 2.0 |
| Holodeck | 27.0 | 0.0 | 0.3 | 9.0 | **10.0** | 8.0 | 7.0 | 27.0 | 0.0 | 4.7 | 7.0 | 6.3 | 4.3 | 4.0 | 35.0 | 0.0 | 12.3 | 5.3 | 4.3 | 4.3 | 3.7 |
| I-Design | 18.7 | 5.3 | 0.0 | 6.0 | 4.5 | 5.8 | 4.3 | 11.7 | 0.0 | 0.0 | 8.0 | 9.0 | 6.8 | 5.4 | 27.7 | 0.0 | 0.0 | 6.2 | 5.2 | 5.2 | 4.0 |
| Ours | **31.0** | 0.0 | 0.0 | 9.0 | 9.0 | 7.7 | **8.0** | **40.0** | 0.0 | 0.0 | 9.0 | 10.0 | 8.0 | 8.7 | **88.0** | 0.0 | 0.0 | 7.3 | 7.0 | 6.5 | 7.3 |

| Method | Waiting Room | | | | | | | Kitchen | | | | | | | Average | | | | | | |
|---|---|---|---|---|---|---|---|---|---|---|---|---|---|---|---|---|---|---|---|---|---|
| | #Obj | #OB | #CN | Real. | Func. | Lay. | Comp. | #Obj | #OB | #CN | Real. | Func. | Lay. | Comp. | #Obj | #OB | #CN | Real. | Func. | Lay. | Comp. |
| LayoutGPT | 6.3 | 0.0 | 0.3 | 6.7 | 5.7 | 6.0 | 4.0 | 7.7 | 1.3 | 1.3 | 5.7 | 6.3 | 4.7 | 3.7 | 7.3 | 0.7 | 0.7 | 6.0 | 6.1 | 5.8 | 3.7 |
| Holodeck | 24.0 | 0.0 | 3.7 | 8.3 | 9.3 | 6.7 | 5.7 | 20.0 | 0.0 | 1.3 | 7.3 | 6.3 | 6.3 | 4.3 | 22.3 | 0.0 | 3.9 | 7.7 | 7.5 | 6.2 | 5.2 |
| I-Design | 10.7 | 0.0 | 0.0 | 6.6 | 6.4 | 5.8 | 4.2 | 11.7 | 0.0 | 0.0 | 6.5 | 6.8 | 5.3 | 3.5 | 14.3 | 0.7 | 0.1 | 7.1 | 7.0 | 6.2 | 4.7 |
| Ours | **25.7** | 0.0 | 0.0 | 9.0 | 10.0 | 8.0 | 7.7 | **34.7** | 0.0 | 0.0 | 9.0 | 9.3 | 7.3 | 7.7 | **36.5** | 0.0 | 0.0 | 8.8 | 9.4 | 7.7 | 8.0 |

**Metrics** For all quantitative evaluations, we evaluate models using physical, visual, and semantic metrics following [3, 7]. For physical evaluation, we report the average number of objects in the scene (#Obj), out-of-boundary objects (#OB), and collided object pairs (#CN) as the main metrics to assess physical plausibility and realism of the scene. For visual and semantic evaluation, we report scores for visual realism (Real.), functionality (Func.), layout correctness (Lay.), and scene completeness (Comp.) as indicators of visual quality and semantic coherence with the user query. Following [7, 8], we use GPT-4 to assess these metrics, providing it with top-down renderings of the generated scenes and the user query as input. To further assess objects' stability in simulation, we evaluate the shift distance in *supplementary*.

## 4.1 Scene Generation for Common Room Types

We provide quantitative evaluation results for the living room and bedroom in Tab. 2. Results show that SCENEWEAVER achieves state-of-the-art results across most metrics, outperforming both data-driven generative models and open-vocabulary models. Notably, Holodeck slightly surpasses SCENEWEAVER in the number of objects (#Obj=32.2). However, we argue that this is primarily due to the inclusion of randomly placed objects, often lacking rationality in object placement. Data-driven methods tend to generate scenes with fewer objects, as their training datasets are largely composed of large furniture items. Consequently, their visual and semantic scores are also lower due to the limited quality and diversity of the training dataset. Interestingly, we observe that data-driven methods outperform LayoutGPT on physical metrics, suggesting that relying solely on LLM-based generation is insufficient for ensuring physical plausibility. In contrast, our LLM-based agentic framework, empowered by reflection and physics-based optimization, achieves zero physical errors, which is comparable to pipelines that enforce hard constraints during optimization (*e.g.*, Holodeck). A qualitative comparison of generated scenes is provided in Fig. 4.

## 4.2 Open-vocabulary Scene Generation

We present quantitative evaluation results in Tab. 3. The results show that SCENEWEAVER significantly outperforms existing open-vocabulary scene generation methods across all eight tested room types. It achieves an average object count of 36.5, notably higher than other approaches, and

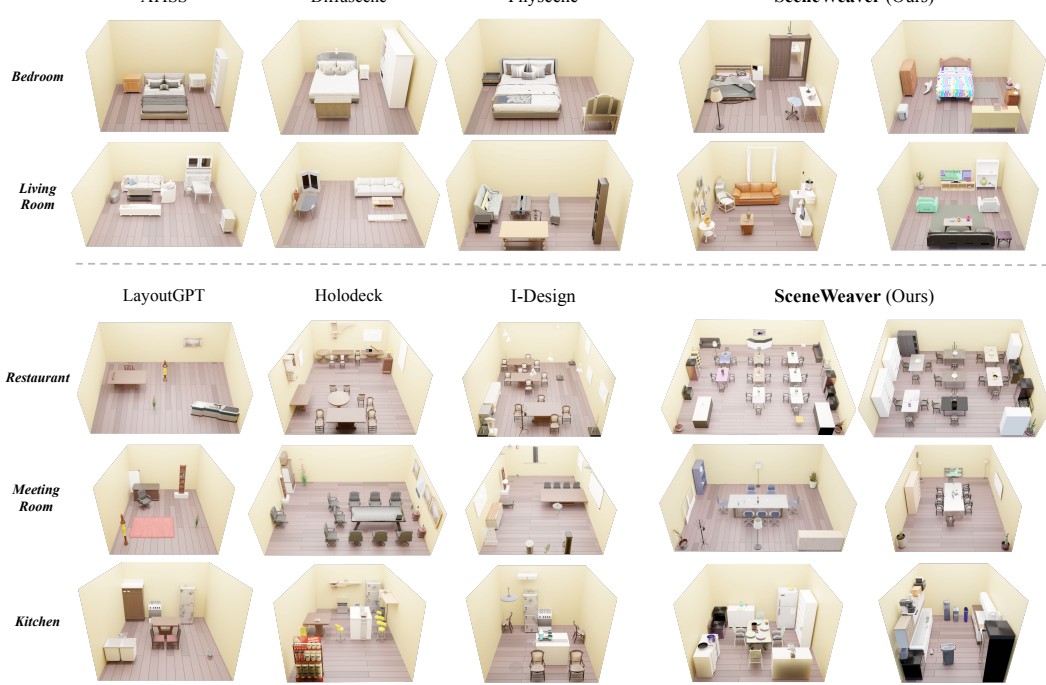

Figure 4: **Qualitative comparison between SCENEWEAVER and existing methods** on both synthesizing common room types and open-vocabulary room types. SCENEWEAVER produces scenes with improved visual realism and finer-grained detail compared to prior methods.

also achieves significantly better visual and semantic scores. More importantly, SCENEWEAVER accomplishes these improvements while strictly satisfying physical constraints (*i.e.*, achieving zero collisions and out-of-boundary violations). This highlights the effectiveness of the reflective planner in both improving semantic coherence with the user query, scene diversity, and in fixing physical implausibilities in the iterative refinement process. We provide qualitative comparisons against other methods in Fig. 4 to further demonstrate the superior visual realism and semantic coherence of scenes generated by SCENEWEAVER. Overall, both quantitative and qualitative metrics confirm that SCENEWEAVER consistently outperforms existing methods on open-vocabulary scene generation, highlighting the effectiveness of our reflective agentic framework.

## 4.3 Additional Analyses

**Ablation on Agent Design** We conduct an ablation study on agent design by evaluating variants of SCENEWEAVER on the average of three kitchen scenes following the open-vocabulary scene generation setting. Specifically, we consider the following variants: 1) removing the reflection module (*w/o* Reflection), 2) removing the physical optimization module (*w/o* Phys. Optim.), and 3) replacing iterative reflection with a single-shot multi-step planning (Multi-step Plan). As shown in Tab. 4, removing the reflection module results in a notable drop in semantic quality, while omitting physical optimization significantly harms physical plausibility. Additionally, compared to the multi-step planning variant, SCENEWEAVER achieves superior visual and semantic performance. This highlights the importance of iterative reflection, as single-pass planning often generates globally inconsistent or locally infeasible layouts by failing to account for context-dependent refinements.

**Effectiveness of Tool Cards** To evaluate the impact of different tool types, we ablate the use of specific subsets from our tool set during scene generation. As shown in Tab. 5, adding or removing particular tool types significantly affects performance across all metrics, demonstrating the importance of tool diversity and validating the design of our standardized Specifically, we observe that modifier tools help align scenes with functional requirements and improve layout coherence, but may reduce object count (16.3 *vs.* 23.0) and completeness (5.0 *vs.* 5.7) by removing redundant items. In contrast, implementer tools excel at enriching scenes with appropriate details, enhancing realism, functionality, and completeness. The full combination of initializer, implementer, and modifier tools yields the highest performance, highlighting the complementary strengths of diverse tools in achieving high-quality 3D scene synthesis.

Table 4: **Ablation on Agent Design.**

| Method | #Obj | #OB | #CN | Real. | Func. | Lay. | Comp. |
|---|---|---|---|---|---|---|---|
| *w/o* Reflection | 25.0 | 0.0 | 0.0 | 8.0 | 8.3 | 6.3 | 6.3 |
| *w/o* Phys. Optim. | 27.3 | 0.7 | 2.0 | 8.3 | 9.3 | 6.7 | 7.7 |
| Multi-step Plan | 29.3 | 0.0 | 0.0 | 8.3 | 7.7 | 7.0 | 7.3 |
| Ours | 34.7 | 0.0 | 0.0 | 9.0 | 9.3 | 7.3 | 7.7 |

Table 5: **Ablation on Effectiveness of Tools.**

| Tools | #Obj | Real. | Func. | Lay. | Comp. |
|---|---|---|---|---|---|
| Init. | 23.0 | 7.7 | 7.0 | 6.0 | 5.7 |
| Init+Modifier | 16.3 | 7.7 | 8.3 | 6.3 | 5.0 |
| Init+Implem. | 34.3 | 8.0 | 8.3 | 6.3 | 7.3 |
| Full | 34.7 | 9.0 | 9.3 | 7.3 | 7.7 |

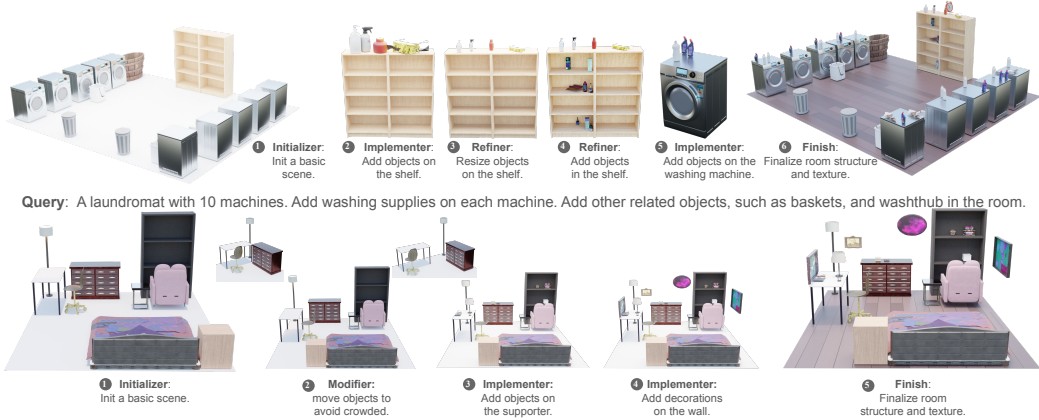

Figure 5: **Iterative refinement in SCENEWEAVER given complex user queries.** SCENEWEAVER progressively incorporates detailed elements specified in the user instruction, demonstrating its ability to iteratively refine and generate high-quality, instruction-aligned 3D scenes (best viewed with zoom-in).

**Iterative Refinement in SCENEWEAVER with Complex Queries**   When presented with complex user instructions, SCENEWEAVER leverages iterative refinement to better follow detailed requirements, particularly in object count, small object placement, and overall scene layout. We provide two qualitative examples of the iterative refinement procedure in Fig. 5 to illustrate this capability.

**Human Study**   To further assess the quality of scenes generated by SCENEWEAVER, we conduct a human study with twenty participants. Each participant evaluates five scenes randomly from the open-vocabulary scene generation setting using the metrics following Sec. 4.2. As shown in Tab. 6, SCENEWEAVER consistently outperforms baseline models across all dimensions. And the Human-LLM alignment is shown in *supplementary*. Additionally, we conduct a pairwise comparison study, where participants indicate their preference and check diversity between scenes (3 for each room type) generated by SCENEWEAVER and baseline methods. Results in Tab. 7 show that SCENEWEAVER is preferred in nearly 85% of cases. These findings underscore the strength of SCENEWEAVER in producing visually and semantically coherent indoor scenes.

Table 6: **Human Evaluation Results.**

| Method | #Obj | Real. | Func. | Lay. | Comp. |
|---|---|---|---|---|---|
| LayoutGPT | 5.94 | 5.83 | 6.21 | 5.26 | 5.99 |
| I-Design | 7.20 | 6.65 | 6.57 | 5.73 | 6.79 |
| Holodeck | 7.86 | 6.70 | 7.35 | 6.67 | 7.45 |
| Ours | **9.30** | **8.80** | **8.85** | **8.55** | **8.98** |

Table 7: **Preference and diversity over other models.**

| Method | *w/* I-Design | *w/* Holodeck | *w/* LayoutGPT |
|---|---|---|---|
| Preference | 94.30% | 91.40% | 87.40% |
| Diversity | 95.60% | 98.90% | 90.00% |

## 5  Conclusion

In this work, we present SCENEWEAVER, a reflective and extensible agentic framework for 3D scene synthesis that integrates diverse scene synthesis paradigms through standardized tool interfaces and iterative feedback-driven refinement. By adopting a reason–act–reflect paradigm, SCENEWEAVER enables an LLM-based planner to dynamically select and invoke appropriate tools, guided by multimodal self-evaluation of physical plausibility, visual realism, and instruction alignment. This closed-loop design allows SCENEWEAVER to effectively decompose and correct complex generation tasks, achieving superior performance across both common and open-vocabulary scene settings. Extensive experiments and human evaluations validate the advantages of our approach in producing high-quality, functionally coherent, and semantically faithful 3D scenes. We believe SCENEWEAVER represents a step toward general-purpose, controllable 3D environment generation, with broad implications for Embodied AI, simulation, and interactive agents.

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

# A Details of the SCENEWEAVER Framework

## A.1 Scene Representation

As mentioned in method, the 3D scene at each step is represented by a combination of 3D layout data and a 2D rendering. The details are illustrated in Fig. A1. On the left, we show a top-down rendering of the scene in Blender, which helps align the visual representation with the coordinate-based layout shown on the right. To enrich the spatial understanding, we mark the image with X, Y, and Z coordinate axes at the coordinate origin and 2D projection coordinates on (x,y) plane to emphasize the spatial position. Each object is further labeled with its 3D bounding box and semantic category to assist the agent in object recognition. We also mark each object with a 3D bounding box and its semantic label to help agent recognize each object. Since visual language models (VLMs) may struggle with spatial reasoning—particularly object orientation—we additionally annotate each bounding box with a directional arrow indicating the object's front. On the right side, the layout encodes each object's semantic category as the key, with its location, rotation, and size as values. We also record relational information for each object, including its parent object and the type of relationship.

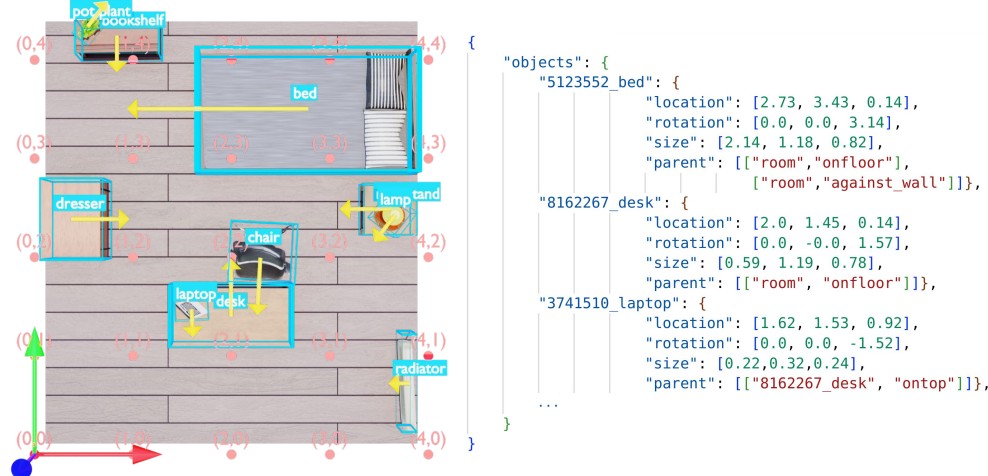

Figure A1: **Example Scene Representation.** To convey both visual and logical information of the current scene, we express the scene data in two representations: 1) a top-down rendered image $\mathcal{I}_t$ (left) with coordinate points, axies-arrow and objects' 3D bounding boxes with labels and direction arrow and 2) the objects' layout $\mathcal{L}_t$ (right) including open-vocabulary category name, location, rotation, size and relation between objects.

## A.2 Self-reflective Planner

We provide the full prompt to the self-reflective planner and feedback mechanism in Tabs. A1 and A2.

## A.3 Implementation of the executor

We take Infinigen as the base code of our executor. And we apply multiple modifications on it to fit our agentic framework. Specifically, we update the code to:

- update iteratively
- interactive in blender in realtime with socker
- fit each tool rather than generating scene procedurelly
- do physical optimization to avoid collision and enhance relation
- add 3D marks and 2D top-down rendering after generation
- …

Table A1: Prompt for planner.

**Prompt for Planner**

**Task description:** You are a scene designer, an expert agent in 3D scene generation and spatial optimization. Your mission is to iteratively design and refine a scene to maximize its realism, accuracy, and controllability, while respecting spatial logic and scene constraints.

**Note:** Given a user prompt, carefully inspect the current configuration and determine the best action to build or enhance the scene structure. You should list all the effective optimization strategy for the next step based solely on geometry, layout relationships, and functional arrangement. You must not focus on style, texture, or aesthetic appearance. To achieve the best results, combine multiple methods over several iterations. Start with a foundational layout and refine it progressively with finer details.

**Available Tools:** {metadata of available tools}

**User demand:** {user_demand}

**Memory of step$_{t-1}$:**

- {planning ideas}
- {tool selection & execution results}
- {scene representation}
- {reflection score & suggestion}

**Plan for step$_t$:**
Based on user needs and current status:

- Clearly explain the execution results of last step and tool.
- According to scene information and evaluation result, check if previous problems have been solved.
- According to evaluation result, which GPT score is the lowest? What physical problem does it have?
- Find the most serious problem to solve.

To solve the problem, list all the appropriate tools that can match the requirement for next step with 0-1 confidence score:

- You should consider the suggestion from previous conversation to score each tool.
- If the same problem has not been solved by last step, you should consider degrade the score of the tool in the last step.
- You should carefully check current scene, and you MUST obey the relation of each object. If there is no previous step, init the scene.
- For complex tasks, you can break down the problem and use different tools step by step to solve it, but you only choose and execute the suitable tool for this step.
- When multiple tools are applicable to solve the user's request, list them with confidence score.

You must choose **one tool** for this step. Clearly explain the expectation and suggest the next steps. If there is no big problem to address, or if only slight improvements can be made, or if further changes could worsen the scene, stop making modifications.

Table A2: Prompt for Verifier.

**Prompt for Verifier**

**Task**  You are given a top-down room render image and the corresponding layout of each object. Your task is to evaluate how well they align with the user's preferences across the four criteria listed below. For each criterion, assign a score from 0 to 10, and provide a brief justification for your rating. Scoring must be strict. If any critical issue is found (such as missing key objects, obvious layout errors, or unrealistic elements), the score should be significantly lowered, even if other aspects are fine.

**Score Guidelines**
- Score 10: Fully meets or exceeds expectations; no major improvements needed.
- Score 5: Partially meets expectations; some obvious flaws exist that limit usefulness.
- Score 0: Completely fails to meet expectations; the aspect is absent, wrong, or contradicts user needs.

**Evaluation Criteria**
1. **Realism**: How realistic the room appears. Ignore texture, lighting, and doors.
   - Good (8-10): The layout (position, rotation, and size) is believable, and common daily objects make the room feel lived-in. Rich of daily furniture and objects.
   - Bad (0-3): Unusual objects or strange placements make the room unrealistic.
   - Note: If object types or combinations defy real-world logic (e.g., bathtubs in bedrooms), score should be below 5.
2. **Functionality**: How well the room supports the intended activities.
   - Good (8-10): Contains the necessary furniture and setup for the specified function.
   - Bad (0-3): Missing key objects or contains mismatched furniture (e.g., no bed in a bedroom).
   - Note: Even one missing critical item should lower the score below 6.
3. **Layout**: Whether the furniture is arranged logically in good pose and aligns with the user's preferences.
   - Good (8-10): Each objects is in reasonable size, neatly placed, objects of the same category are well aglined, relationships are reasonable (e.g., chairs face desks), sufficient space exists for walking, and orientations must be correct.
   - Bad (0-3): Floating objects, crowded floor, abnormal size, objects with collision, incorrect orientation, or large items placed oddly (e.g., sofa not against the wall). Large empty space. Blocker in front of furniture.
   - Note: If the room has layout issues that affect use, it should not score above 5.
4. **Completion**: How complete and finished the room feels.
   - Good (8-10):All necessary large and small items are present. Has rich details. Each shelf has multiple objects inside. Each supporter (e.g. table, desk, and shelf) has small objects on it. Empty area is less than 50%. The room feels done.
   - Bad (0-3): Room is sparse or empty, lacks decor or key elements.
   - Note: If more than 50% of the room is blank or lack detail, score under 5.

**User demand**   {user_demand}
**Rendered Image**   {rendered_image $\mathcal{I}_t$}
**Room layout**   {layout $\mathcal{L}_t$}
**Results**   Return the results in the following JSON format, the comment should be short:

```
{ "realism": {
    "grade": your grade as int,
    "comment": "Your comment and suggestion."
  },
  "functionality": {...},
  "layout": {...},
  "completion": {...}}
```

## A.4  Tool Cards

We provide detailed prompts to all tools in Tabs. A8–A17. And we list the implementation details of each tool in  Tab. A3. During the planning process, the agent is provided with a brief description of

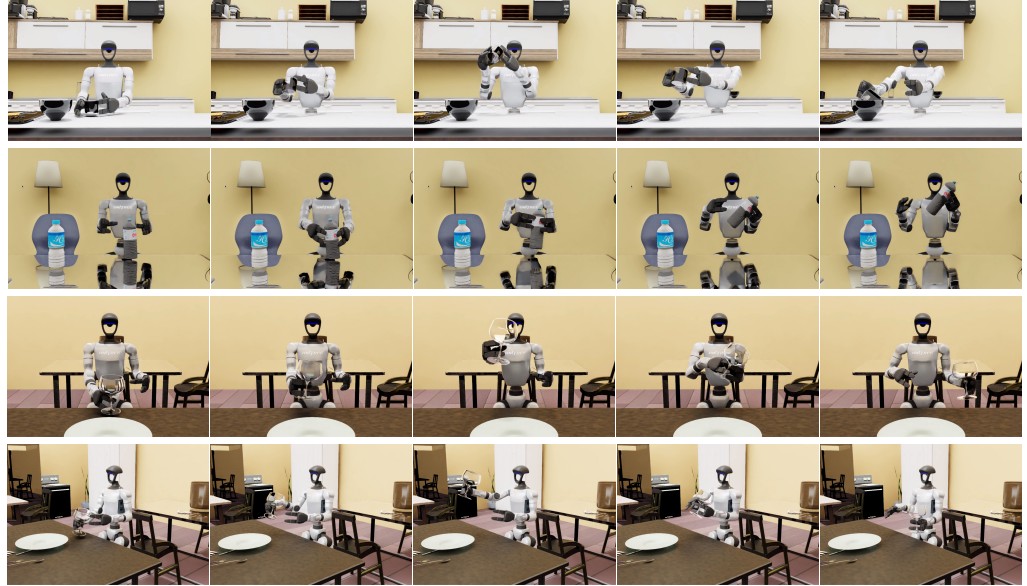

Figure A2: **Robot interacts with the scene generated by SCENEWEAVER in simulation.** The first three rows show the sequences of interaction in the front view in three different scenes including kitchen, meeting room and restaurant. And the last row shows the side view of the third row. Note the system keeps different materials, such as table in the meeting room has transparent and reflective material.

each tool and uses the function calling to choose a single tool for each step. Then it will run the tool itself and go through the executor to update the scene.

## A.5 Physics-aware Executor

Referring to Infinigen, the relation types here include two aspects.

1. **Relation between the object and room:**
   - `<against_wall>`: the object's back faces to the wall, and stands very close or exactly on the wall.
   - `<side_against_wall>`: the object's side (left, right, or front) faces to the wall, and stands very close.
   - `<on_floor>`: the object stands on the ground.

2. **Relation between two objects:**
   - `<front_against>`: the child object's front faces to the parent object, and stands very close, such as chair and dining table.
   - `<front_to_front>`: the child object's front faces to the parent object's front, and stands very close, such as chair and desk, coffee table and sofa.
   - `<leftright_to_leftright>`: the child object's left or right faces to the parent object's left or right, and stands very close.
   - `<side_by_side>`: the child object's side (left, right , or front) faces to the parent object's side (left, right , or front), and stands very close.
   - `<back_to_back>`: the child object's back faces to the parent object's back, and stands very close.
   - `<on_top>`: the child object is placed on the top of the parent object, such as monitor and desk, vase and table.
   - `<inside>`: the child object is placed inside the parent object, such as book inside shelf.

## A.6 How to choose assets dataset

In this project, we choose different asset according to the usage of tool. You can also choose any of them to suit your own requirements.

Table A3: **Tool Overview and Contributions.**

| Tool Name | Role | Use of Existing Works | Our Contribution |
|---|---|---|---|
| Init MetaScenes | Init scene with Real2Sim dataset | MetaScenes | Choose data & convert format |
| Init PhyScene | Init scene with pre-trained model | PhyScene / DiffuScene / ATISS | Choose data & convert format |
| Init GPT | Init scene with GPT | LLM | Prompt engineering |
| Add ACDC | Add tabletop objects visually | Stable Diffusion & ACDC | Significant changes on digital cousin |
| Add GPT | Add objects with GPT | VLM | Prompt engineering |
| Add Crowd | Add crowded layout | VLM & Infinigen | Utilize Infinigen rules & design module |
| Remove Object | Remove inappropriate Object | VLM | Prompt engineering |
| Add Relation | Add relations to objects | VLM & Infinigen | Prompt engineering & Utilize Infinigen relations |
| Update Rotation | Fix rotation problems | VLM | Prompt engineering |
| Update Size | Rescale objects | VLM | Prompt engineering |
| Update Layout | Update improper layouts | VLM | Prompt engineering |

**MetaScenes**: For tool using **Dataset** such as MetaScenes, we employ its assets directly, since each scene contains several assets with delicated mesh and layout information.

**3D FUTURE**: For tool using **Model** such as PhyScene / DiffuScene / ATISS, we employ 3D FUTURE, since the model is trained on this dataset.

**Infinigen**: For other tools, we use Infinigen's asset generation code to generate standard assets in common categories, such as bed, sofa, and plate. The asset will be generated in a delicated rule procedure in the scene generation process.

**Objaverse**: For those categories that are not supported by Infinigen, such as clock, laptop, and washing machine, we employ open-vocabulary Objaverse dataset.

# B  Experiments

## B.1  Additional Experimental Details

The maximum number of steps is set to 10. However, the procedure may terminate earlier if the intermediate results already meet the user's requirements with a high score. The reflection module determines whether to continue optimizing or stop.

For asset retrieval, we gather resources from available tools when possible. For instance, when using tools based on data-driven methods, we adopt 3D-FUTURE assets. If no assets are provided, we first rely on the Infinigen generator to produce standard assets following predefined rules. For more open-vocabulary assets, we refer to OpenShape to retrieve objects from Objaverse. In cases where assets lack a unified initial pose, we calculate their minimum bounding rectangles to identify four side candidates, then prompt GPT to annotate the front-facing direction. GPT achieves a high success rate in identifying the front side of commonly known objects, though it may fail for more complex or ambiguous cases. For the ablation study, we focus on the kitchen room type and generate three scenes for the experimental setting.

## B.2  Additional Scene Generation Results

We show more visualization results of SCENEWEAVER in Fig. A3. The results of restaurant, garage, and gym confirm that SCENEWEAVER is able to arrange multiple objects neatly when the number of

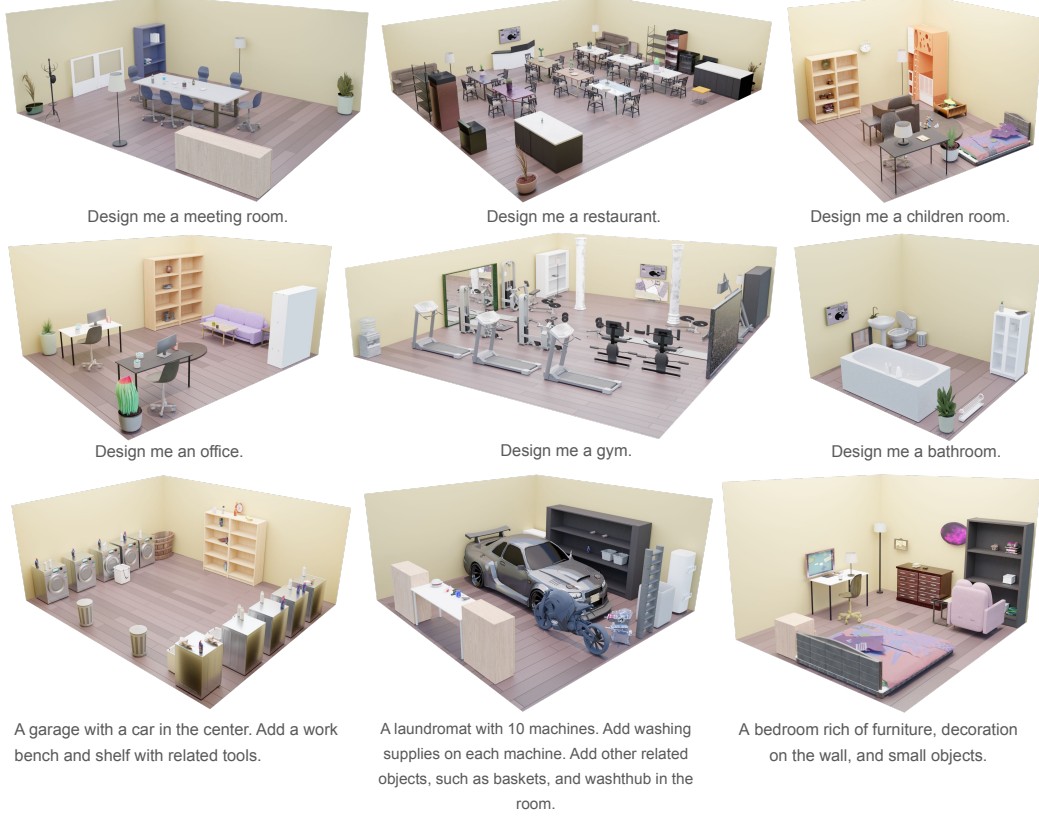

Figure A3: **More visualization examples of generated scenes.**

the same category is more than three. Cabinet in the bathroom contains objects inside, such as a roll of paper, since it has supporting surface in the plane inside. Shelves are equipped with related objects inside (basket in garage and towel in gym). The third row shows some detailed results of complex user queries.

## B.3 Simulation in Isaac Sim

We export the generated scenes as USD files and load them into Isaac Sim for physical simulation and interactive tasks. Through Apple Vision Pro, we remotely control a Unitree G1 humanoid robot to perform object interactions within these virtual environments. As demonstrated in Fig. A2 and our supplementary video, the system supports diverse interaction scenarios across multiple scenes: the first three rows showcase interaction sequences from a front-view perspective, while the last row provides a side-view analysis of the third scene. This pipeline offers three key advantages for embodied AI applications:

- High-fidelity simulation with preserved textures and geometric details.
- Robust physical interactions guaranteed by collision-free and boundary-constrained object placement.
- Task-aligned scene layouts that adapt to diverse EAI requirements through controllable synthesis.

With the combination of these features, we believe SCENEWEAVER enables reliable sim-to-real transfer for robotic manipulation tasks while maintaining visual and functional realism.

## B.4 User Study

We invited twenty participants to evaluate the scenes generated by SCENEWEAVER. All participants were volunteers without compensation. We invite them to assess the scenes in two settings.

Table A4: **Score alignment.** To further confirm the stability of LLM's judgement and human study, we check the alignment between different users as well as LLM

| Method | Realism ↑ | Functionality ↑ | Layout ↑ | Completion ↑ |
|---|---|---|---|---|
| User-User | 0.43 | 0.42 | 0.45 | 0.40 |
| User-LLM | 0.46 | 0.45 | 0.48 | 0.55 |

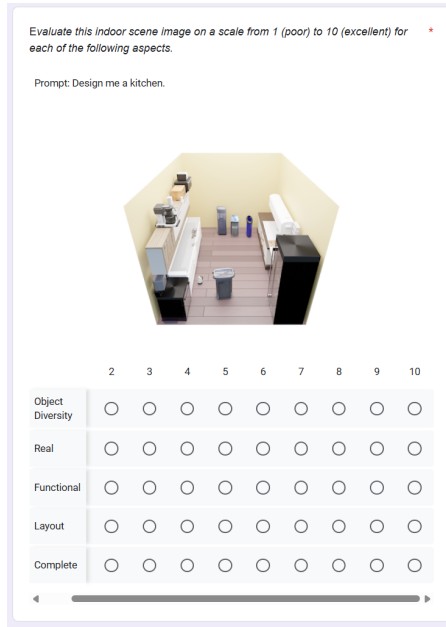

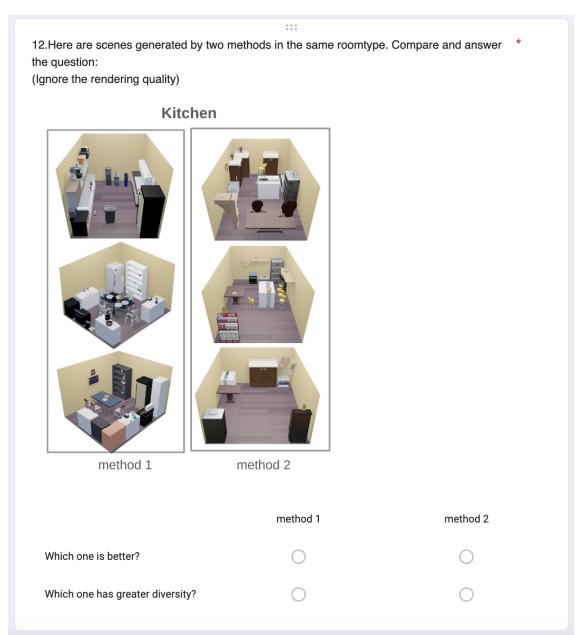

(a) The first setting to score each scene.  (b) The second setting for pairwise comparison.

Figure A4: **Example of user study in different settings.**

In the first setting, we randomly collected 100 scenes generated by three baseline methods and SCENEWEAVER. Each volunteer was randomly assigned 20 scenes along with their corresponding prompts. Participants were asked to rate each scene on a scale from 1 to 10 using the same five metrics described in the experiment. Note that physical metrics such as collision count (#CN) and out-of-boundary objects (#OB) were excluded from this human evaluation, as they are difficult to assess by eye. For each of the five remaining metrics, we provided a guiding sentence to help participants make consistent and informed judgments. Finally, we aggregated the ratings from all participants and computed the average score for each metric.

In the second setting, we conducted a pairwise comparison study, shown in Fig. A4b. For each baseline method, we selected five set of scenes, where each set includes three scenes generated by SCENEWEAVER and three scenes generated by the baseline method under the same prompt. We increased the generated scene number from 1 to 3 for each method in order to 1) reduce individual variance and 2) check diversity of different generation system. We asked the participants to choose:

- Which method do you prefer?
- Which method has greater diversity?

We collected votes from all participants and calculated the average preference between SCENEWEAVER and each baseline method. Results in Tab. 7 shows our methods has greater diversity while also stand for higher preference, surpassing other methods to a large degree.

## B.5  Human-LLM Alignment.

Note the evaluation results of human evaluation and LLM score can not be exactly the same, thus a slight difference between them is a normal phenomenom. To further confirm the stability of these justification methods, we check the alignment between different users and LLM. We convert the users' and LLM's scores on different scenes to rankings and calculate Kendall's Tau for the 4

Table A5: **Shift in simulation.** We assess object stability in simulation in Isaac Sim.

| Method | >0.1m ↓ | >0.01m ↓ | Average Shift ↓ |
|---|---|---|---|
| ATISS | 35.4% | 51.4% | 0.356 |
| DiffuScene | 26.2% | 39.3% | 0.190 |
| PhyScene | 9.7% | 19.6% | 0.069 |
| LayoutGPT | 39.2% | 52.8% | 0.477 |
| IDesign | 5.0% | 11.5% | 0.041 |
| Holodeck | 17.6% | 42.5% | 0.113 |
| Ours | **1.0%** | **10.37%** | **0.011** |

Table A6: **An example of metric improvement during iteration.**

| Step | Tool | Realism ↑ | Functionality ↑ | Layout ↑ | Completion ↑ |
|---|---|---|---|---|---|
| 1 | Initializer | 6 | 6 | 5 | 4 |
| 2 | Implementer | 7 | 6 | 5 | 4 |
| 3 | Refiner | 8 | 7 | 6 | 6 |
| 4 | Refiner | 8 | 7 | 8 | 6 |
| 5 | Implementer | 8 | 7 | 8 | 8 |

metrics, shown in Tab. A4. The results show strong user-user agreement and user-LLM agreement on different metrics. And we find that user-LLM has higher alignment scores than user-user, which indicates that LLM is more stable than user in evaluation. These alignment results also demonstrate the effectiveness of using LLM for justification in Tab. 3 of the main paper.

## B.6 Physical Stability

To assess object stability in simulation, we measure:

- Shift thresholds: percentage of objects moving >0.1m or >0.01m in 3s
- Average displacement: Mean shift distance (meters)

As shown in Tab. A5, the lowest shift of our methods confirms our great performance on the original physical metrics #OB and #CN. Above all, our generated scene remains the most stable in simulation although with the largest object number.

## B.7 Metric improvement during iteration.

The overall is self-adaptive rather than coarse-to-fine object placement. The agent in our pipeline will find the biggest problem in each iteration according to the reflection score and choose related tool to solve the problem. For example, the teaser in Fig. 1 serves as an obvious example. Here we show the metric scores after executing each step in Tab. A6.

The agent chooses the lowest score to improve. Specifically, step 1 has the lowest score in completion, which is 4. Then in the step 2, the agent chooses the implementer to add objects in the shelf and improve the completion score from 4 to 6. Now the lowest score is the layout, which is 5, owing to the illogical location of bathroom sink. So in step 3, it uses refiner to remove the bathroom sink, improving the layout score from 5 to 6. So now the lowest score belongs to both layout and completion. In step 4, the agent focuses on the crowding area and decides to use refiner to rearrange the dining tables to ensure adequate space for movement and clear walkways. Then the score of layout becomes 8. The lowest score comes to the completion again. So the agent choose implementer again to add objects on each table to improve the completion score.

The scores of realism and functionality will also be improved while we focus on optimizing other metrics. And the agent can fix the problem caused by previous steps, such as removing the bathroom sink generated by the previous step. And it can utilize the same tool multiple times to improve the quality until it satisfies the demand.

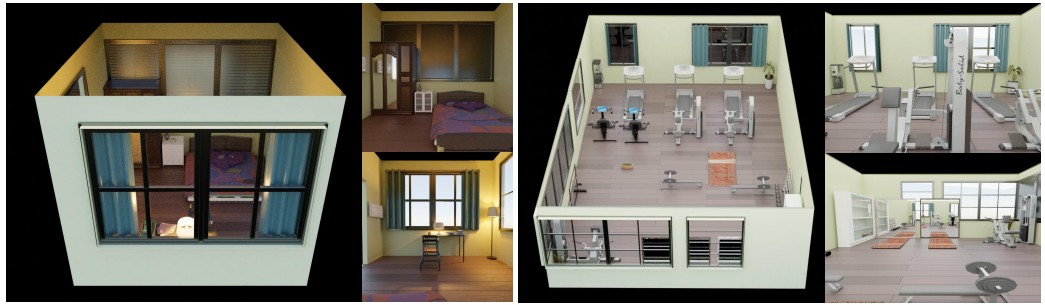

Figure A5: **Adding room structure including door and windows.** We show two samples of scene generation with room structures. Each sample includes views from both outside and inside.

Table A7: **Improvement on the scenes generated by baseline models.**

| Method | #Obj ↑ | Realism ↑ | Functionality ↑ | Layout ↑ | Completion ↑ |
|---|---|---|---|---|---|
| PhyScene | 5 | 7 | 8 | 6 | 5 |
| PhyScene + Ours | 11 | 9 | 10 | 8 | 9 |
| LayoutGPT | 6 | 7 | 8 | 6 | 5 |
| LayoutGPT + Ours | 20 | 9 | 9 | 8 | 10 |

### B.8 Impact of a single tool

Tab. 5 shows that the addition of different types of tools (Initializer, Refiner, Implementor) greatly improves the results. To further assess the effect of a single tool, we observe that without the "Update Rotation tool, the model is less sensitive to invalid rotations, reducing the "layout" score. Meanwhile, the "Add Crowd" tool lets supporters be densely filled with child objects (e.g., a shelf packed with books), which improves "realism". If there are multiple tools in similar function, adding/removing a single tool will not make a big difference. And if the tool has a negative effect, the agent with the memory and reflection module will recognize it during iterations and avoid using the tool.

### B.9 Room Structure & Single Room Scale

In the main paper, we remove windows and doors to simplify the generation process, though our method can handle such structures and update the scene accordingly. Here we show some examples in Fig. A5.

For room scale, we focus on single rooms to improve scene quality as prompted by the user. The process can be repeated to handle multiple rooms. We could also generate multi-room scenes at once with some coordinate transformation.

### B.10 Refine scenes generated by baseline methods

Here we refine scenes generated by two baseline methods, PhyScene and LayoutGPT. (IDesign and Holodeck will take longer time in format conversion.) Results show our method could greatly improve the baseline method. Actually the baseline method serves as an initializer tool in this setting, and we can convert each baseline method to a new tool and merge them into the extensible tool cards.

## C  Miscellaneous

### C.1  Claims in Tab. 1

Here we clarify criteria dicussed in Tab. 1.

**Real**    Defined as realistic layouts considering human daily habits.

**Large-scale**    Methods able to generate infinite scene variations.

**Accurate**  Methods with well-defined rules ensuring explicit object relations and strict placement.

For example, Infinigen's realism relies on simple rule, texture and rendering, but layout issues remain: (1) monitors may face walls, as placement rules lack direction awareness; (2) random sampling without spatial priors can scatter desks. Therefore, we do not consider Infinigen "real".

## C.2   Broader Impact

Our work focuses on 3D scene synthesis, aiming to generate physically interactable environments based on complex, user-specific instructions. A key application lies in the development of embodied artificial intelligence, where such synthesized scenes can be used to train agents across diverse tasks. Furthermore, the overall architecture of SCENEWEAVER is grounded in recent LLM-based tool-use agent frameworks, positioning it to inspire future agentic systems tailored to specific use cases. This includes the design of task-specialized components such as system prompts and interaction protocols for enhanced application-specific performance. At present, we do not anticipate any immediate negative societal impacts resulting from SCENEWEAVER.

## C.3   Resources used

All reported experiments are conducted on a machine equipped with an NVIDIA GeForce RTX 4090 GPU. To generate a scene, the time consumption ranges from minutes to hours, depending on the iteration number, chosen tools, and crowded status. We use Blender 3.6 to record and render the scene.

## C.4   Time Consumption

The time consumption is a bit longer due to several reasons. First, different method takes different time. For example, the data-driven tool is fast, since the process is simple and the model is trained in advance. While the 2D guided tool, such as ACDC, is slower, since the process is complex and included several procedures including 2D segmentation, 3D reconstruction, assets matching, and pose optimization. Another reason is that we add physical optimization in the executor to ensure the physical plausibility in the geometric level, while previous work only consider the bounding box level. The third reason is because we take several steps to develop a scene rather than a single step.

The average time for a single iteration is 8.6 minutes, and the average time for generating a complete scene is 64 (min:35, max:130) minutes. Simpler and smaller room need less generation time. Here are some choices to reduce the time cost:

- reduce the usage frequency of the physical optimization
- remove some time-consuming tools, such as digital cousins. Although the results will be influenced, it will not cost too much drop.
- reduce the iteration number.

## C.5   API Cost

The average iteration is about 7. The average API cost is about 0.5 dollar per scene. The relatively higher cost come from well-designed agentic framework, making the great results on both physical and visual & semantic metrics is irreplaceable by other methods. More diffusion steps or longer Infinigen procedure will not change the difference.

Moreover, here are some choices to reduce the cost by:

- simpler requirements and smaller room size.
- reduce the usage frequency of the physical optimization.
- remove some time-consuming tools or replace with other effective tools.
- reduce the iteration number.

Note reducing iteration number will not affect the final result too much, since the scene achieves high score quickly in the first few iterations. The rest iterations aim to fix corner cases and improve the

score slightly. For example, the GPT score for bedroom is 8.6, 9.3, 7.5, 7 when we limited the steps to 3, still better than the baseline methods.

# D    Limitation

## D.1    Unrealistic Scaling & Placement for Functional Items

A small ratio of objects may look unreal in the scale and placement. This problem comes from the non-standard open-vocabulary dataset in two points. On one side, the object in open-vocabulary dataset such as Objaverse has no standard size, for example, a toy can be 10 meters in length. Thus it is necessary to resize those assets to eliminate the gap and fit the realness. On the other side, the front direction of the asset is not provided in the dataset, thus we utilize VLM to predict the front direction, which we find with high success rate, but it still failed in some cases since sometimes the front direction of the asset can not be defined uniformly, such as the elliptical bike, our expected front direction is mistaken as the side direction, leading to unreasonable orientation and size. How to balance the standardization and generality for the open-vocabulary dataset is an unsolved problem, but it still provides a feasible and promising view for general scene synthesis.

## D.2    Validity of LLM's Judgement

To evaluate the generation system, human study stands for a fair choice, but it is very expensive to scale. The LLM provides a cheaper, faster and more stable (as discussed in  Sec. B.5) choice to evaluate. Taking LLM for both generation and validation seems conflict, while here we consider it as acceptable, since the LLM score is easy to acquire during generation and does not result in overfitting. Previous work GPTEval3D [36] have also proved that VLM can serve as a Human-Aligned Evaluator for Text-to-3D Generation. SceneCraft [37] also utilizes VLM to critic and revise the scene with Blender script.

We acknowledge that LLMs are imperfect proxies for human judgment. Alternative scaffolds (e.g., rule-based metrics, heuristic checks, or hybrid human-AI pipelines)) exist but offer no clear advantage in alignment or scalability. In the absence of better accessible evaluators, we treat LLM feedback as a practical interim solution. And we remain hopeful that future research will yield more reliable, scalable, and human-aligned evaluation paradigms.

Based on the current situation, we emphasize that, given an accessible LLM model to guide the generation system, the key point is how to utilize the guidance and find a good way to improve the scene quality. Using GPT's judgments during iteration also demonstrates the unique ability our reflective agentic framework. Other baseline methods with fixed pipelines and limited tools lack the ability to adjust results based on evaluation scores or LLM feedback, while ours makes it possible to iteratively refine scenes through feedback-driven planning with modular tools.

Table A8: Metadata of Initializer: Real2Sim - MetaScene.

**Initializer: Real2Sim - MetaScene: Metadata**

**Description**   Load the most related scene from the Real2Sim indoor scene dataset MetaScenes as the basic scene. Ideal for generating foundational layouts for common room types.
**Supported Room Types:**   living room, dining room, bedroom, bathroom, kitchen, hotel, office, laundry room, and classroom.
**Use Case 1**   Create a foundational layout.
**Strengths**   Provides a ready-made layout based on real-world data. Rich of details.
**Weaknesses**   Fixed layout, need to modify with other methods to meet user demand.
**Input**   Roomtype, Ideas to init the scene.

Table A9: Metadata of Initializer: Model - PhyScene.

**Initializer: Model - PhyScene: Metadata**

**Description**   Using PhyScene, a neural network, to generate a scene as the basic scene. The model is trained on the 3D Front indoor dataset.
**Supported Room Types**   Living room, bedroom, and dining room.
**Use Case 1**   Create a foundational layout.
**Strengths**   Room is clean and tidy. Assets in good quality.
**Weaknesses**   Fixed layout with less details.
**Input**   Roomtype, Ideas to init the scene.

Table A10: Metadata of Initializer: LLM - GPT.

**Initializer: LLM - GPT: Metadata**

**Description**   Using GPT to generate the foundamental scene.
**Supported Room Types**   any room type.
**Use Case 1**   Create an accurate and foundational layout.
**Strengths**   Align well with user demand. More details. Highly versatile and capable of generating scenes for any room type and complex user requirement. Flexible with respect to room design and customization.
**Weaknesses**   Less spatial rationality. May not be as real as data-driven and Real2Sim methods.
**Input**   Roomtype, Ideas to init the scene.

Table A11: Metadata of Implementer: 2D Guided - ACDC.

**Implementer: 2D Guided - ACDC: Metadata**

**Description**   Using image generation and 3D reconstruction to add additional objects into the current scene.
**Use Case 1**   Add a group of small objects on the top of an empty and large furniture, such as a table, cabinet, and desk when there is nothing on its top.
**Strengths**   Real. Excellent for adding a group of objects with inter-relations on the top of a large furniture.(e.g., enriching a tabletop), such as adding (laptop,mouse,keyboard) set on the desk and (plate,spoon,food) set on the dining table. Accurate in rotation.
**Weaknesses**   Can not add objects on the wall, ground, or ceiling. Can not add objects inside a container, such as objects in the shelf. Can not add objects when there is already something on the top.
**Input**   Ideas to add objects.

Table A12: Metadata of Implementer: LLM - GPT.

**Implementer: Implementer: LLM - GPT: Metadata**

**Description**   Using GPT to add additional objects into the current scene.
**Use Case 1**   Add large objects in the current scene.
**Use Case 2**   Add 1-2 small objects on the top of small supporting furniture, such as nightstand and cabinet, when there is enough space. (e.g., add a cup on the nightstand).
**Use Case 3**   Add several small objects on the top of large supporting furniture, such as dining table and desk, when there is enough space. (e.g., add daily tableware on the dining table).
**Use Case 4**   Add several small objects inside the large furniture. (e.g., add books in the shelf).
**Use Case 5**   Add functional objects or decorations on the wall. (e.g., add painting, mirror, and TV on the wall).
**Strengths**   The location is accurate. Can add objects inside a container, such as objects in the shelf.
**Weaknesses**   The rotation of asset is not always accurate. Relation between small objects is not clear. Can not modify objects in the current scene. Can not add objects on the ceiling.
**Input**   Ideas to add objects.

Table A13: Metadata of Refiner: LLM - Remove Object.

**Refiner: LLM - Remove Object: Metadata**

**Description**   Remove objects with GPT. Works with all room types.
**Use Case 1**   Remove redundant and unnecessary objects when the scene is crowded or when there are too many objects. (e.g., eliminate a table in the corner)
**Use Case 2**   Remove objects that does not belongs to this roomtype. (e.g., eliminate the bed in the dining room)
**Use Case 3**   Remove objects when the collision/outside problem has not been solved for several attempts by other tools. (e.g., eliminate the object outside the room)
**Use Case 4**   Remove small objects (usually with collision or outside the supporting surface) when their supporter or container has no enough space to support them. (e.g., eliminate some small objects or when the nightstand is overloaded)
**Strengths**   Excels at removing specific objects. Can solve collison and crowded problems directly.
**Weaknesses**   Can not add objects or replace objects. You must use this method carefully to avoid mistaken deletion.
**Input**   Ideas to remove objects.

Table A14: Metadata of Refiner: LLM&Rule - Add Relation. The relation types are introduced in Sec. A.5.

**Refiner: LLM&Rule - Add Relation: Metadata**

**Description**   Add explicit relation between objects in the current scene according to the layout. Sometimes the relation is encoded in the layout coordinate rather than represented explicitly, making it difficult to manage. Explicit relations will make the scene more tidy.
**Note**: Each object can have only one parent object (except for the room). Do not add relation between small objects.
The optional relations between objects are {relation_types}.
**Use Case 1**   Add explicit relation between large objects, according to the layout, to make the scene better-organized.
**Use Case 2**   Add new relation between large objects, make the scene better-organized.
**Use Case 3**   Add againts_wall relation to large objects, make the objects stand against wall.
**Use Case 4**   Add floating small objects on/in a large object.
**Strengths**   Can add relation between objects, make the scene tidy and well-organized quickly.
**Weaknesses**   Can not fix the layout problem, such as placing the object into the right place accurately.
**Input**   Ideas to add relation.

Table A15: Metadata of Refiner: VLM - Update Rotation.

**Refiner: VLM - Update Rotation: Metadata**

**Description**   Adjust object rotations with GPT to optimize room layout.
**Use Case 1**   Fix incorrect object orientations, such as a bed facing the wall or a chair turned away from a desk.
**Use Case 2**   Improve spatial organization by aligning objects more naturally with the room structure or usage context (e.g., rotate a sofa to face a TV or a chair to face a table).
**Strengths**   Helps improve the visual and functional coherence of a room. Can automatically identify misaligned items and suggest better orientations based on typical room usage.
**Weaknesses**   Does not move, add, or remove objects. Only focus on rotation.
**Input**   Ideas to update rotation.

Table A16: Metadata of Refiner: LLM - Update Size.

**Refiner: LLM - Update Size: Metadata**

**Description**   Modify Object Sizes with GPT. Best suited for significant size adjustments rather than minor refinements.
**Use Case 1**   Resizing objects with abnormal proportions (e.g., an object on a table that is over one meter tall).
**Use Case 2**   Scaling objects to meet functional requirements (e.g., enlarging a table when a larger one is needed).
**Strengths**   Effective at adjusting specific object sizes.
**Weaknesses**   Cannot modify overall room dimensions. Should only be used when necessary due to potential scene inconsistencies.
**Input**   Ideas to update size.

Table A17: Metadata of Refiner: LLM - Update Layout.

**Refiner: LLM - Update Layout: Metadata**

**Description**   Modify layout with GPT.
**Use Case 1**   Adjust objects' placement when the objects are not well-placed.
**Use Case 2**   Change objects' scale when the size does not match the requirement.
**Strengths**   Excels at modifying specific objects. This method is not recommended for slight layout adjustments. It is better suited for major changes when necessary.
**Weaknesses**   Can not solve all the problem when the room is crowded. Poor in modify rotation. May lack precision and occasionally overlook details. Can not obey the current relation, such as move object away from the wall when the object is against wall. Can not add objects.
**Input**   Ideas to update layout.

