# OpenReview forum: "SceneWeaver: All-in-One 3D Scene Synthesis with an Extensible and Self-Reflective Agent"
_NeurIPS.cc/2025/Conference — NeurIPS 2025 poster_

### Official Review · Reviewer_dp6k · 2025-06-16

**Clarity:** 3
**Significance:** 2
**Originality:** 2
**Rating:** 4
**Confidence:** 3

**Summary:**

The paper introduces a language-model-based agentic framework to generate both common and open-vocabulary room types. It dynamically selects tools and iteratively refines the scene based on feedback from language models, and it achieves better results than baselines on both human evaluation and language-model evaluation.

**Questions:**

- Can the author provide real “iterative” examples where the model rebuilds part of the scene to improve each of the metrics: visual realism (Real.), functionality (Func.), layout correctness (Lay.), and scene completeness? Otherwise, it is hard to assess the significance of the approach.
- Can the author provide experiments on using the proposed framework to refine scenes generated by baseline methods, so that the scenes under comparison look mostly similar but improved?

**Ethical Concerns:**

["NO or VERY MINOR ethics concerns only"]

**Final Justification:**

Most of my concerns are resolved, therefore I increase my rating from 3 to 4

**Limitations:**

yes

**Quality:**

2

**Strengths And Weaknesses:**

Strengths:
- The framework unifies a lot of existing scene synthesis methods and improves on them.
- The paper is easy to follow with a lot of figures to help understand.

Weaknesses
- My overall impression is that the pipeline can be done through simple optimization without language models. From figure 5, it is hard to see the “iterative” part (maybe except for the move of the table). It seems the pipeline is just about coarse to fine object placement, similar to Infinigen.
- In figure 4, because these scenes are all different, a naive approach could just be to let the model sample a lot of different scenes and (automatically) cherry-pick one. Because the other approaches do not do cherry-picking, it is easy to outperform them. So it would be best to have more restricted comparisons.
- The evaluation metric is mostly language-model-based. So it is likely “the blind leading the blind.” Though there are human evaluations, they are limited by the number of participants.
- In table 1, the criteria might be too strict, for example, Infinigen is classified as not “Real”.
- The controllability of the method is limited, at least in the experiment. The prompt is only the room type.

---

> ### Author Rebuttal · Authors · 2025-07-31
>
> Thank you for the valuable feedback and suggestions.
> ### Q1: More participants in human evaluations.
> We improve the user number to 20 and increase the evaluated scene number to 200, each of them is shown  40 samples.  We show them the same instructions used as prompt in the LLM reflector for the participant to evaluate the scene carefully.
> We statistic the results into the following table, which indicates similar results with the previously shown Table 3 in the main paper.
>
> | Method      | Real.  $\uparrow$   | Func.   $\uparrow$  | Lay.   $\uparrow$   | Comp.    $\uparrow$ |
> |-------------|-----------|-----------|-----------|-----------|
> | LayoutGPT   | 5.83     | 6.21    | 5.26    | 5.99    |
> | I-Design    | 6.65     | 6.57     | 5.73     | 6.79     |
> | Holodeck    | 6.70     | 7.35     | 6.67     | 7.45     |
> | ours        | 8.80    | 8.85     | 8.55     | 8.98     |
>
> ### Q2: GPT for both generation and validation
> We understand your concern. To evaluate the generation system, human study stands for a fair choice, but it is very expensive to scale. The LLM provides a cheaper, faster and more stable (as shown in the user-LLM alignment) choice to evaluate. Taking LLM for both generation and validation seems conflict, while here we consider it as acceptable, since the LLM score is easy to acquire during generation and does not result in overfitting. We could also add other scanffoldings, but they may not provide a better evaluation. We can also apply GPT to filter the results of other methods such as ATISS, but they will still returns lower score due to the limitation of the method itself. For example, ATISS will be limited to the data distribution of the training dataset, trapped in limited categories of large furniture and simple layout of only 3 room types. Therefore, we emphasize that, given an accessable LLM model to guide the generation system, the key point is how to utilize the guidance and find a good way to improve the scene quality.
>
> ### Q3: More controllable prompt
> We have shown some samples of complex prompts in Figure A2 in the supplementary material. It shows the controllability of objects’ category (basket, washtub, etc.), number (”A laundromat with 10 machines”), position (”a car in the center”), and precise components (decoration on the wall, small objects).
> We will release more results in the final version and web page, since the rebuttal can not show additional images and external links.
>
> ### Q4: Use SceneWeaver to refine scenes generated by baseline methods
> That is a good idea. Due to time constraint in rebuttal, here we refine scenes generated by two baseline methods, PhyScene and LayoutGPT. (IDesign and Holodeck will take longer time in format conversion.)
>
> | Method      | #Obj | Realism | Functionality | Layout | Completion |
> |-------------|------|---------|---------------|--------|-------------|
> | PhyScene    | 5    | 7       | 8             | 6      | 5           |
> | +ours   | 11   | 9   | 10        | 8  | 9       |
> | LayoutGPT   | 6    | 7       | 8             | 6      | 5           |
> | +ours   | 20   | 9   | 9         | 8  | 10      |
>
> Results shows our method could greatly improve the baseline method.
> Actually the baseline method serves as a initializer tool in this setting, and we can convert each baseline method to a new tool and merge them into the extensible tool cards.
>
> ### Q5: “Iterative” examples to improve each of the metrics
> The overall is self-adaptive rather than coarse to fine object placement. The agent in our pipeline will find the biggest problem in each iteration according to the reflection score and choose related tool to solve the problem. For example, the teaser serves as an obvious example.
> Here we show the metric score after executing each step:
> | Step | Tool         | Realism | Functionality | Layout | Completion |
> |------|----------------|---------|---------------|--------|------------|
> | 1    | Initializer       | 6       | 6             | 5      | **4**      |
> | 2    | Implementer        | 7       | 6             | **5**      | 6          |
> | 3    | Modifier  | 8       | 7             |**6**    | 6          |
> | 4    | Modifier         | 8       | 7             | 8      | **6**          |
> | 5    | Implementer        | 8       | 7             | 8      | 8          |
>
> The agent will choose the lowest score to improve. Specifically, the step 1 has lowest score in completion, which is 4. Then in the step 2, the agent chooses the implementer to add objects in the shelf and improve the completion score from 4 to 6. Now the lowest score is the layout, which is 5, owing to the illogical location of bathroom sink. So in step 3, it uses modifier to remove the bathroom sink, improving the layout score from 5 to 6. So now the lowest score consist of both layout and completion. In step 4,  the agent focuses on the crowding area and decide to use modifier to rearrange the dining tables to ensure adequate space for movement and clear walkways. Then the score of layout becomes 8. The lowest score come to the completion again. So the agent choose implementer again to add objects on each table to improve the completion score.
>
> You can see the realism and  functionality will also be impoved while we focus on optimizing other metrics. And the agent can fix the problem caused by previous steps, such as removing the bathroom sink generated in previous step. And it can utilize the same tool for multiple times to improve the quality untill it fit the demand.
>
> ### Q6: Sampling strategy
> The Sampes in Figure 4 and Figure A2 is randomly choose from the generated scenes. Figure 4 also show  2 samples to show variance of our method. And the results in the tables are statistic on all the generated results with the related prompts.
>
> And we further implement a user study to confirm the stability and diverisity of our method.
> We increase the generated scene (randomly chosen) number from 1 to 3 for each method in the second comparison setting in order to 1) reduce individual variance and 2) check diversity of different generation system.
> We ask the participants to choose:
>
> - Which method do you prefer?
> - Which method has greater diversity?
>
> | Method   |   w/ LayoutGPT | w/ I-Design | w/ Holodeck |
> |-----------|--------|---------------|-----------------|
> | Preference    | 94.30% | 91.40%        | 87.40%          |
> | Diversity | 95.60% | 98.90%        | 90.00%          |
>
> Results shows our methods has greater diversity while also stand for higher preference, showing the stability of our method.

---

> > ### Comment · Reviewer_dp6k · 2025-08-05
> >
> > Thanks for your detailed response. I think most of my concerns are resolved (except those mentioned by DJFe), and I will very likely raise my rating to 4.

---

> > > ### Author Response · Authors · 2025-08-06
> > >
> > > Thank you for your encouraging feedback.

---

### Official Review · Reviewer_DJFe · 2025-07-01

**Clarity:** 3
**Significance:** 3
**Originality:** 2
**Rating:** 3
**Confidence:** 4

**Summary:**

The paper presents SceneWeaver, an LLM-based system which integrates many asset libraries and generation tools to perform open-vocabulary language-conditioned generation of indoor 3D environments. SceneWeaver uses multimodal language models to iteratively suggest improvements ("reflection") and choose tools to use to add assets, add physical constraints, or correct errors detected by the LLM.

**Questions:**

Are the existing qualitative results random and non-selected? What is the variance in visual quality if I evaluated random seeds 0...5 with the exact same bedroom prompt?

What is the technical contribution of the tool implementations and standardized tool cards? How do the tool cards expand on a normal LLM-tool-use setup e.g. describing the tools to the openai api? The presence/non-presence of tools is ablated, but _standardized and extensibility_ is listed as a contribution (L74).

What parts of the "executor" (L202) are built by the authors vs taken from existing tools?

I would appreciate any discussion or further justification of Tab2/3/6 quantitative results (see above). In my view, the LLM-as-judge results should be removed from the paper.

What is the mean/min/max SceneWeaver cost in terms of CPU-hours, GPU-hours and API credits? What are the costs for some of the examples scenes in the main paper? What are the realism metrics if SceneWeaver is restricted to the runtime of the baselines? How do we know that the cost-realism tradeoff of SceneWeaver is favourable for downstream tasks (robotics/games) compared to baselines?

I believe the work is promising and would argue for acceptance if the authors address my concerns on evaluation and cost.

**Ethical Concerns:**

["NO or VERY MINOR ethics concerns only"]

**Final Justification:**

Table 2&3's evaluation setup is potentially flawed. The metrics are graded by GPT-4, but SceneWeaver's method also prompts GPT-4 to give iterative feedback on these same metrics during generation. Accessing the validation score in this way is invalid. The authors provided new empirical evidence that this does not hugely bias the results, but the conceptual issue has not been acknowledged, and I believe this creates a bad precedent or may mislead readers.

Related work makes unclear claims via Tab 1 check/cross marks that previous work is not "accurate" or "physically plausible", which are not binary or objective. This is not major, but was not acknowledged in rebuttal after being raised twice by me and once by DP6K.

The authors significantly strengthened their other quantitative results during rebuttal, and overall the qualitative and quantitative results are compelling. I maintain a borderline reject on the basis that the Tab 2/3 issues are potentially serious, but Tab2/3 could instead just be removed.

**Limitations:**

Runtime is listed as a limitation (which is good to report) but requires further details discussion (see above).

No "potential negative impacts" are mentioned but I do not see this as significant.

**Paper Formatting Concerns:**

Some llm-oriented terms should be defined or made more concrete. E.g. "reasoning signal" and especially "reflective agentic framework".

Sec4.3 "Effectiveness of toolcards" has an incomplete sentence "demonstrating the
283 importance of tool diversity and validating the design of our standardized [???]" (brackets[] mine).

**Quality:**

2

**Strengths And Weaknesses:**

Qualitative results:
- strength: the qualitative results provided are compelling, especially offices and gyms which are challenging (due to objects not attached to walls, e.g. grid of desks or treadmills) and are not supported in prior tools e.g. ProcThor, Infinigen
- weakness: lacking a larger controlled qualitative sample. ATISS and Infinigen both provide 30+ random qualitative samples in their supplement. The paper would be stronger if it included such a sample, ideally seeds=0,1,2 for each language prompt (so we could have seen the variance due to random seed) and with outputs for other methods for the same prompt or scene category.

Tool use:
- strength: the work integrates an impressive variety of object and scene synthesis tools, which I believe takes nontrivial engineering effort
- weakness: it is unclear in the main paper which tools use what underlying woreks. paragraph L199 should be significantly expanded in the main paper to make the technical implementation of these tools clear.
	- The work uses assets from Objaverse, 3D-Future, Infinigen "depending on the tool" (L204) but doesnt state how these choices are made (even in A.3). L75 states tools are used "based on their respective strengths" - how does the system know or encode these strengths?
	- It is unclear how object resizing works - are the assets scaled up/down (unrealistically?) or does it use infinigen parameters?
	- A.3. specifies the arrangement constraint system, but it extremely closely resembles Infinigen's solve_state.json.
	- The paper main paper could specify this all carefully (ideally a table with columns "tool name", "role", "our contribution", "use of existing works") and expand paragraph L199.

Minor issue: Table 2/3's `#OBJ, #OB and #CN` metrics are gameable
- These metrics are easily saturatable (many works score 0.0 error), and in theory can be satisfied by a weak baseline (e.g. a shape-packing algorithm that places objects without intersection but with no regard for realism or even gravity).
- `#OB` and `#CN` are necessary but not sufficient for use in robotics sims. e.g. objects could be floating or topple over in the sim, but wouldnt violate these metrics. Some objects in the demo video would topple in this way.
- A better metric would be "what is the sum of total pose changes when we initialize a simulation and let objects settle?". This would subsume `#OB` and `#CN` (since intersections cause large movements), but would also measure other bad phenomena (object topples over, object drops to the floor). https://arxiv.org/abs/2405.20510v1 and similar use metrics along these lines I believe.

Table 2/3's use of GPT-4 as a judge of "Realism", "Functionality" and "Layout" is invalid for SceneWeaver, (even despite that "LLMs as judge" is now a common paradigm for other work).
-  The SceneWeaver reflection system uses GPT-4 (the exact judge model, mentioned on L178) to actively iterate against the exact "real/func/layout" validation scores (confirmed in Appendix Table A2) . If the method accesses the validation scores during generation, then in my view they are invalid as validation scores.  This is similar to reporting the train loss of a deep neural network, or training on a validation set. If the comparison setup of Tab2/3 says this is fair game, ATISS should be allowed to generate 1000 samples and return whichever GPT-4 says is best, before then being judged by GPT-4.
- A premise of the work is that LLMs need significant scaffolding (e.g. SceneWeaver) in order to correct deficiencies in spatial reasoning. However, the work then uses GPT-4 with minimal scaffolding as quantitative evaluation.
- A more minor issue: Using natural language prompts as the definition of "functionality" is imprecise. Functionality could have a precise geometric definition, e.g. what affordances are preserved (% of sofas/seating that can be accessed from the front, or % of percent of ovens/cabinets that have space to open without obstruction). These are directly important for downstream tasks. I recognize this is a minor and very stringent criticism however, as most prior works do not evaluate this.

The authors do provide a human user study of 5 participants, which partially addresses my concerns in Tab2/3. In my view, this needs to serve as the main evaluation of the system. Therefore, it would benefit from more participants and a stronger study design to reduce variance. Specifically the user study would ideally:
- (1) clearly define what evaluators should look for in terms of "real", "functional" and "layout" (currently there are no instructions to participants what these mean - see appendix A7)
- (2) provide an evaluation of layout which specifically says to ignore asset / rendering quality (similar to Infinigen-Indoors)
- (3) show multiple outputs for each generation system, to evaluate which _distribution_ of results is better (evaluating individual samples will be high variance)
Overall, the human user study shows much more minor gains over Holodeck than the GPT-4-as-judge study, which lends credence to the idea that LLMs-as-judges may not be valid for this paper.

The work does provides insufficient description and argument on the cost of the system. AppendixC-L76 states it is "minutes to hours" but this could mean 24 hours. The work uses GPT-4, so it also has a precise dollar-cost of APIs used per scene. CPU cost, GPU cost, and API cost should each be reported as a mean, min and max, or even reported for individual qualitative results, or means reported in Tab2/3 for each method if available. Along with this, I would expect an argument that the cost is worthwhile for the improved results (e.g. "+X minutes is acceptable for embodied sims because it can be amortized over Y minutes of sim usage"). If the cost is very very high, it would be rigorous to also evaluate against baselines on an equal-cost setting (fewer reflection steps for SceneWeaver, or maybe increase DiffuScene denoising steps / run Infinigen with very large annealing steps).

In summary, I believe the work does produce impressive scene arrangements, but would appreciate controlled and more numerous qualitative results. My major concerns are that (1) Table2/3 metrics should not be presented as primary evidence of the work's usefulness and (2) the work does not report detailed cost, does not weigh cost in any comparisons, and does not argue for the acceptability of its cost.

"Quality" Justification: technical soundness - LLM-as-judge is not technically sound, requires additional evaluation (qualitative and quantitative). some claims in Tab1 are potentially excessive ("large-scale", "real"(ism?), "accurate" - all are subjective and may apply to baselines)

"Clarity" Justification: good except for description of how underlying works are used.

"Significance": potentially very useful as a robotics / sim tool. usefulness depends on cost.

"Originality" Justification: LLM "reflection" or tool use are not original, and i don't believe their application to scene arrangement is highly original, unless I am misunderstanding details of the "tool cards".

---

> ### Author Rebuttal · Authors · 2025-07-31
>
> Thank you for the valuable feedback and suggestions.
> ### Q1: User Study
> **User Number and Sample Size**
>
> We enhanced the study's robustness by scaling to **20** participants evaluating 200 scenes (40 samples each), using **identical LLM reflector's instruction of metrics** to ensure consistent evaluation criteria. Similar to Infinigen-Indoors' approach of focusing solely on furniture layout's realness without texture and rendering, our original instructions (shown in Table A2) also emphasize to **evaluate realness by excluding texture and lighting considerations.** This unified evaluation framework, combined with our Objaverse-based pipeline, minimizes platform/asset biases. ）The results remain consistent with Table 3's findings.
>
> | Method| Real.  $\uparrow$   | Func.   $\uparrow$  | Lay.   $\uparrow$   | Comp.    $\uparrow$ |
> |-|-|-|-|-|
> | LayoutGPT   | 5.83     | 6.21    | 5.26    | 5.99    |
> | I-Design    | 6.65     | 6.57     | 5.73     | 6.79  |
> | Holodeck    | 6.70     | 7.35     | 6.67     | 7.45 |
> | ours | 8.80    | 8.85     | 8.55     | 8.98     |
>
> ### Score Alignment and Validity of Evaluation
> Given that human and LLM evaluations naturally differ slightly, we assessed their consistency by converting scores to rankings and computing Kendall's Tau across all four metrics. This confirms the stability of LLM's justification.
>
> | Alignment | Real.   $\uparrow$  | Func. $\uparrow$    | Lay.  $\uparrow$    | Comp. $\uparrow$    |
> |--|-|-|-|--|
> | User-User | 0.43| 0.42|0.45|0.40|
> | User-LLM | 0.46| 0.45|0.48|0.55|
>
> Results demonstrate strong agreement between users and LLM across metrics, with higher user-LLM alignment than user-user (τ=0.55 vs 0.40 on Completion metric), confirming LLM's superior evaluation stability. These findings validate our LLM justification approach in Tables 2/3.
>
> ### Diversity and Preference
>
> We increase the scene number from 1 to 3 for each method in the second comparison setting in order to 1) reduce individual variance and 2) check diversity of different methods.
> We ask the participants to choose:
>
> - Which method do you prefer?
> - Which method has greater diversity?
>
> | Method | w/ LayoutGPT | w/ I-Design | w/ Holodeck |
> |-|-|-|-|
> | Preference    | 94.30% | 91.40%  | 87.40% |
> | Diversity | 95.60% | 98.90% | 90.00% |
>
> Results shows our methods has greater diversity while also stand for higher preference, surpassing other methods in a large degree.
>
> ### Q2: GPT for both generation and validation is invalid. need more scanffolding.
>
> We understand your concern. To evaluate the generation system, human study stands for a fair choice, but it is very expensive to scale. The LLM provides a cheaper, faster and more stable (as shown in the user-LLM alignment) choice to evaluate. Taking LLM for both generation and validation seems conflict, while here we consider it as acceptable, since the LLM score is easy to acquire during generation and does not result in overfitting. We could also add other scanffoldings, but they may not provide a better evaluation.
>
> Using GPT to filter the results of  ATISS will still returns lower score due to the limitation of the method itself, trapped in data distribution of the training dataset with limited categories and room types. Therefore,  given an accessable LLM model to guide the generation system, we emphasize how to utilize the guidance and find a good way to improve the scene quality.
>
> ### Q3: “Functionality” could have a geometric definition such as accessibility and interactivity.
> This is a good suggestion. The “functionality” you mentioned is related to precise calculation for the physical interactivity, which is hard for LLM to reply. Here we refer to PhyScene and SceneEval and add additional physical metric `accessibility`.
> Due to the time limits, here we randomly choose 3 bedrooms of each method and check the percentage of large furniture (such as bed, wardrobe, and nightstand) that can be reached in the front.
>
> | Method    | Accessibility |
> |--|--|
> | LayoutGPT | 77%   |
> | IDesign   | 80% |
> | Holodeck  | 82%  |
> | ours  | 82%   |
>
> Our methods shows high accessibility since the agent is able to check the walkable area. For example, the step 4 in Figure 1 shows the agent has recognized the crowding area and rearranged the dining tables to ensure adequate space for movement and clear walkways. Note the Holodeck shows relatively high accessibility. That’s because its room has larger size than other methods and objects are closed to the wall.
>
> ### Q4:  Physical Metrics
> **Validity of `#OB` and `#CN`**
>
> Here we do not use shape packing or floating placement for higher physical score. The size of assets is decided by each tool in reasonable process. And each object has as least one relation (contact closely with out collision) with the room (floor/wall) or supporting objects (table/shelf/…), ensuring that nothing is floating in the room. Thus table 2/3's `#Obj`, `#OB` and `#CN` metrics are solid.
>
> **New Metric：Shift in simulation**
>
> Thanks for your suggestion.   To assess object stability in simulation, we measured:
>
> - Shift thresholds: percentage of objects moving >0.1m or >0.01m in 3s
> - Average displacement: Mean shift distance (meters)
>
> | Method  | >0.1m |>0.01m | Avg Shift  |
> |-|---|--|---|
> | ATISS   | 35.40% | 51.40%  | 0.356 |
> | DiffuScene  | 26.20%  | 39.30%  | 0.190  |
> | Physcene    | 9.70%  | 19.60%  | 0.069  |
> | LayoutGPT   | 39.20%     | 52.80%  | 0.477   |
> | IDesign  | 5.00%   | 11.50% | 0.041 |
> | Holodeck | 17.60% | 42.50%  | 0.113  |
> | Ours | 1.00%  | 10.37% | 0.011   |
>
> The lowest shift of our mthods verify our great performance on the original physical metrics `#OB` and `#CN`. Obove all, our generated scene remains the most stable in simulation with the largest object number, which further confirm the great ability of our method’s  physically suitable position.
>
> ### Q5: Cost
> The **average time** for a single iteration is 8.6 minutes, and the average time for generating a complete scene is 64 (min:35, max:130) minutes. The average iteration is about 7.
> The average **API cost** is about 0.5 dollar per scene. And Holodeck report its cost as 0.2 per scene. IDesign and LayoutGPT does not release the cost.
> Since other baseline does not provide **CPU&GPU cost**, we think the comparison might not be precise.
>
> The relatively higher cost come from well-designed agentic framework, making the great results on both physical and visual & semantic metrics is irreplaceable by other methods. More diffusion steps or longer infinigen procedure will not change the difference.
>
> Moreover, here are some choices to reduce the cost by:
> - simpler requirements and smaller room size.
> - reduce the usage frequency of the physical optimization.
> - remove some time-consuming tools or replace with other effective tools.
> - reduce the iteration number.
>
> Note reducing iteration number will not affect the final result too much, since the scene achieves high score quickly in the first few iterations. The rest iterations aim to fix corner cases and improve the score slightly. For example, the GPT score for bedroom is 8.6, 9.3, 7.5, 7 when we limited the steps to 3, still better than the baseline methods.
>
> ### Q6: Implementation of the tools
> We list the implementation of each tool.
>
> | Tool Type   | Tool Name    | Role  | Use of Existing Works  | Our Contribution  |
> |-|-|-|-|-|
> | Initializer | Init MetaScenes | Init scene with Real2Sim dataset  | MetaScenes                  | Choose data + convert format         |
> |   | Init PhyScene   | Init scene with pretrained model  | PhyScene/DiffuScene/ATISS   | Choose data + convert format         |
> |   | Init GPT        | Init scene with LLM    | LLM(GPT)    | Prompt engineering |
> | Implementer | Add ACDC        | Add tabletop objects visually     | Stable Diffusion + ACDC     | Significant changes on digital cousin|
> | | Add GPT  | Add objects with LLM   | VLM | Prompt engineering    |
> |  | LLM+Rule        | Add crowded layout                | LLM+Infinigen  | Utilize Infinigen rules + design module |
> | Refiner | Remove Object    | Remove invalid objects            | LLM(GPT)   | Prompt engineering |
> |  | Add Relation | Add relations to objects  | VLM(GPT) + Infinigen  | Utilize Infinigen relations + prompt engineering |
> |   | Update Rotation  | Fix rotation problems  | VLM(GPT) | Prompt engineering  |
> |  | Update Size      | Rescale objects  | LLM(GPT)  | Prompt engineering   |
> |  | Update Layout    | Update improper layouts | VLM(GPT)  | Prompt engineering   |
>
> During the planning process, the agent is provided with brief description of each tool, shown in A3-A13 in supplementary material and use the function calling to choose a single tool for each step. Then it will run the tool itself and go through the executor to update the scene.
>
> ### Q7: Implementation of the executor
> Note we not only contribute on the tool itself, but also modify the infinigen to :
> - update iteratively
> - interactive in blender in realtime with socker
> - fit each tools rather than generate scene procedurelly.
> - Physical optimization
> - Add 3D marks and 2D top-down rendering
> - …
>
>
> ### Q8: How to choose assets dataset
> In this project, we choose different asset according to the usage of tool. And you can also choose any of them to fit your own requirements.
> - MetaScenes: For tool using Dataset such as MetaScenes, we employ its assets directly, since each scene contains several assets with delicated mesh and layout information.
> - 3D FUTURE: For tool using Model such as PhyScene/DiffuScene/ATISS, we employ 3D FUTURE, since the model is trained on this dataset.
> - Infinigen: For other tools, we use Infinigen's asset generation code to generate standard assets in common categories, such as bed, sofa, and plate. The asset will be generated in a delicated rule procedure in the scene generation process.
> - Objaverse: For those catrgories that are not supported by Infinigen, such as clock, laptop, and washing machine, we employ open-vocabulary Objaverse dataset.

---

> > ### Comment · Reviewer_DJFe · 2025-08-04
> >
> > Thanks for your response. The shift in simulation results and other new quantitative results are appreciated. However, I believe some concerns have not been addressed:
> >
> > \textbf{Using GPT-4 for both generation "reflection" and quantitative validation}
> >
> > I find the written argument for using LLMs as quantitative evaluation unconvincing. The authors state "here we consider it as acceptable... does not result in overfitting", but this seems hard to prove. The authors cite FirePlace and LayoutVLM (in response to kZsN) but to me this misses the core issue. FirePlace and LayoutVLM do not query GPT-4's percieved criticisms of the layout during generation, whereas Sceneweaver does. Sceneweaver can "reflect" on the generation until GPT-4 is satisfied, before being graded by GPT-4.
> >
> > Similarly, dp6k and myself both raise a hypothetical. In my wording: one could just repeatedly sample ATISS until GPT-4 gives the scene a high score. The authors do not address this, they say it would be "trapped in data distribution of the training dataset", but even still the score will still be better if we take the max over n trials using GPT-4.
> >
> > The users vs llms alignment table is helpful. It would help to detail exactly how this is computed and whether the results are statistically significant. It is not intuitive that the User-LLM alignment is higher than User-User - if the humans are noisy then it should be hard for other measurements to be particularly correlated.
> >
> > The VLM-judged results are not necessarily a dealbreaker, as there are other quantitative and qualitative results that seems strong. But I do worry they are unfair to other work, or may misdirect readers or future work. I don't think the empirical alignment between the two has fully resolved this (what if the results are just generally noisy? what trend does it set?), and I don't think the authors have addressed the core issue.
> >
> > \textbf{table 1 claims}
> > dp6k and myself both criticized the checkmark/cross claims in Table 1 and this has not been addressed. In what sense is ProcThor/Infinigen not "real", when all other works are also using/generating synthetic datasets? Why are many prior works not "accurate" or "physically-plausible" ? These seem to be a subjective continuous spectrum, not a yes-no answer, and broadly should be revised.

---

> > > ### Comment · Reviewer_dp6k · 2025-08-05
> > >
> > > Hi DJFe,
> > > Thanks for sharing the thoughts, I agree that the author’s written argument for using LLMs as quantitative evaluation and the address for “repeatedly sample ATISS“  are unconvincing. But I have asked for “Using SceneWeaver to refine scenes generated by baseline methods” and the author has shown its great improvement. I think this can be used as better evidence to dismiss the “repeatedly sample” hypothetical (though still not a completely fair comparison). What do you think?

---

> ### Author Response · Authors · 2025-08-06
> **Repeatedly sample ATISS**
>
> We thank reviewer DJFe for the response.
>
> Here we list the detailed scores of 10 scenes generated by ATISS, which can be regarded as **sampling ATISS ten times** （10 is also the maximum iteration number of the agent in SceneWeaver).
>
> We calculate the best score of each metric, which can be approximately regraded as **taking the max over 10 trails using GPT-4**.
>
> |            | Realism $\uparrow$ | Functionality$\uparrow$  | Layout $\uparrow$ | Completion $\uparrow$ | #Obj $\uparrow$ | #OB $\downarrow$ | #CN $\downarrow$ |
> |------------|---------|---------------|--------|------------|------|-----|-----|
> | ATISS_1    | 8       | 8             | 7      | 5          | 5    | 1   | 1   |
> | ATISS_2    | 7       | 6             | 8      | 4          | 4    | 0   | 0   |
> | ATISS_3    | 9       | 8             | 7      | 5          | 5    | 1   | 1   |
> | ATISS_4    | 6       | 5             | 4      | 3          | 3    | 0   | 0   |
> | ATISS_5    | 9       | 8             | 8      | 7          | 5    | 1   | 2   |
> | ATISS_6    | 7       | 7             | 5      | 4          | 5    | 0   | 1   |
> | ATISS_7    | 6       | 5             | 7      | 3          | 2    | 0   | 0   |
> | ATISS_8    | 8       | 8             | 7      | 4          | 4    | 2   | 1   |
> | ATISS_9    | 7       | 8             | 7      | 4          | 3    | 0   | 0   |
> | ATISS_10   | 7       | 8             | 6      | 3          | 3    | 0   | 0   |
> | **ATISS mean** | 7.4  | 7.1           | 6.6    | 4.2        | 3.9  | 0.5 | 0.6 |
> | **ATISS best** | 9    | 8             | 8      | 7          | 5    | 0   | 0   |
> | **Ours (SceneWeaver)** | **9.2** | **9.8**     | **8.4**    | **9.4**        | **14**   |**0**   | **0**   |
>
>
> Results shows the **best score of ATISS still falls short compared to ours**.
>
>
> Moreover, due to the limited dataset, ATISS is in lack of **scene diversity**, taking the best over multiple trails may make it worse. Meanwhile, ATISS has no small or open-vocabulary objects, the #Obj is low, and the shift in simulation is high.
>
>
> Also, according to the refinement results of PhyScene (more details shown in Q4 for Reviewer dp6k), we believe **our method can further refine ATISS's result to a higher score**, which can not be achieved by repeated sampling and max-selection with GPT-4.  (Here PhyScene and ATISS are similar methods)
>
>
> | Method      |  Realism | Functionality | Layout | Completion |#Obj |
> |-------------|------|---------|--------------|--------|------------|
> | PhyScene    |7       | 8            | 6      | 5          | 5    |
> | +ours       | 9       | 10           | 8      | 9          |11   |
>
> Above all, although using GPT to filter ATISS can make some improvement, there still exists a big gap between ATISS and our method.

---

> > ### Comment · Reviewer_DJFe · 2025-08-09
> >
> > These empirical results still do not address the issue in Table 2&3's evaluation setup at a conceptual level.
> >
> > To recap the issue: Tab 2&3 use GPT-4's opinion to grade SceneWeaver's output. But SceneWeaver also iterates against GPT-4's opinion as part of its method (not present in most/all baselines). SceneWeaver's iterative "verifier" even specifically prompts GPT-4 to critique Realism, Functionality and Layout, which are the same validation metrics graded by GPT-4 in Tab 2/3.
> >
> > The authors provide empirical evidence to justify that SceneWeaver is likely not overfitting to GPT-4's opinion in practice. The most recent results show that a hypothetical posed by myself&DP6K (resampling ATISS) partially closes the gap, without surpassing their method. But the conceptual issue remains. If a model was trained on the test set, this would be a serious issue, no matter how much empirical evidence shows that it is not significantly overfit.
> >
> > There has also been no response regarding unclear claims in related work Table 1. These have been raised twice by me above, and also once by DP6K, although I recognize they may have been missed due to large volume of other questions.
> >
> > I would like to ensure these issues are discussed among reviewers/AC. However I am going to maintain my score as is for the timebeing. I believe the Tab2/3 issue could be serious, however these tables are also not essential to the paper or could be addressed by acknowledging in writing.

---

### Official Review · Reviewer_Er4e · 2025-07-01

**Clarity:** 3
**Significance:** 2
**Originality:** 3
**Rating:** 4
**Confidence:** 4

**Summary:**

This paper introduces a system, Sceneweaver, for generating simulable single room scenes from open language prompts with "reason -> act -> reflect" strategy. The system primarily leverages a central planning MLLM with mixtures of LLM,MLLM,VLM and some model-based algorithms as "tools" to initialize and iteratively refine a scene. These tools scaffold the iterative process and provide clear direction to the central planner about what to refine and how to do so. The tools are broken into "initializers", "microscene implementers", and "detail refiners" with a few of each type implemented, though more could be added. The authors validate their results against SotA systems with quantitative analysis and small scale user study. They also demonstrate a potential robotics simulation application by loading the generated scenes into a physics simulator for teleoperated manipulation.

**Questions:**

- We know there isn't contact between objects from the metrics, but what about excess space? If simulation is run on the scenes, how much shifting occurs in the states? Are object placements stable?
- How do we iterate on a system like this? Since the system relies so heavily on the mixture of tools and their implementations, what metrics would we use to evaluate its improvement when we add or remove individual tools?
  - Can we evaluate the impact/effectiveness of a single tool in isolation?
- "the secret sauce" - looks like we're missing details about the implementation of the tools themselves. Beyond the tool cards and scattered discussion we're lacking concreteness on these details.
  - e.g. how exactly is the
- We see "functional diversity" as a focus for this project, but we don't see any analysis of diversity specifically mentioned. Is this feasible to add for a small prompt domain? How much visual and content diversity can one achieve with a fairly precise prompt?

**Ethical Concerns:**

["NO or VERY MINOR ethics concerns only"]

**Final Justification:**

My understanding of the work and my opinion of the strengths and weaknesses have not changed, nor has my rating which was provided under the assumption of my understanding. Overall I am satisfied with the additional information provided by the authors to answer my questions and confirm where my criticisms should be mentioned as additional limitations and avenues for future work.  I feel the final version of the exposition will be improved by the increased transparency and additional details.

**Limitations:**

Primary limitation claimed by the authors is the generation time and complexity.

Likely, other limitations related more to the specific mixture of tool ingredients present in this iteration rather than a fundamental system limitation given that tools can be expanded. However, they could mention:
- limited complexity of relationships - mouse+keyboard+monitor is a good start, but real robotics tasks will require much more and "well aligned" content
- hallucinations of realism - e.g. "squashing" of functional items like exercise equipment or blocking of articulable items like drawers.
- clearly no structural elements are considered (windows, doors, stairs) and rooms are always rectangular
- functional access does not seem to be validated: can an agent reach the furniture or objects to use them?

**Paper Formatting Concerns:**

Typos:

LN 105: isdesigned
Table A5: "Using GPT to generate the 'foundamental' scene."
Table 2 column headers "Physcis" and "Funtc."
Figure A2: "washthub"
Figure A1: "axies-arrow"
Table A2: 3. Layout: "aglined"
Table A7: "Remove objects that does not belongs to this roomtype."
LN 119: "reason-act-react" should be "a reason–act–reflect"?

**Quality:**

3

**Strengths And Weaknesses:**

I like the approach. Iterative language model planning with tool use is a clear strategy and the authors executed on this approach to improve on SotA metrics. I do think some more details could be included and more limitations could be addressed, particularly on the diversity of results and anti-physical content. Also, if robotics is a primary target application it would be good to try the system in task contexts to validate that it generalizes beyond simple open prompts. A user study was nice to see and results were clear, but more than 5 participants would be nice. Overall I'm recommending a weak accept but would consider stronger support with some more details added.

Pros:
- open-language input
- focus on physically interactive scenes targets robotics simulation well
- closed loop iterative refinement  vs. monolithic
- pre-defined "tools" inject the hand-designed or model-based organized elements which MLLMs can leverage more openly
    - the "secret sauce" is the combination and implementation of these tools
- human study to validate the results
- Captures hierarchical support relationships between objects
- extensible with more tools
- supports multiple input modalities (2D guided vs. data-driven)
- as models improve, so should results
- because tools have language feedback, it can be understandable compared to quantitative optimization algorithms (e.g. based on statistical priors)
- object relationships are considered explicitly (e.g. chairs facing the desk)

Cons:
- single room scale
- lots of prompt engineering
- unrealistic scaling/placement for functional items (e.g. the gym layout with vertically stacked treadmills and squashed elliptical bikes.
    - At the end of the day, function dictates parameter limits. You can't scale an elliptical machine and a chair must be large enough to fit a person. These hallucinations limit the effectiveness of the final scenes.
- Large generation times (up to hours) limits utility in practice
- minor (out of scope): no articulations or considerations - e.g. drawers and drawers need to open
- 5 users study with 20 scenes each seems a bit lightweight
- No task-based specifications (yet?) - interesting to see that robotics simulation was the target here, but task doesn't manifest in examples. I suppose a prompt could include the task like "generate a kitchen scene with dishes next to the sink for dish washing", but it would be interesting to see that demonstrated if so.

---

> ### Author Rebuttal · Authors · 2025-07-31
>
> Thank you for the valuable feedback and suggestions.
> ### Q1: More participants in user study.
>
> We enhanced the study's robustness by scaling to 20 participants evaluating 200 scenes (40 samples each), using identical LLM reflector instructions to ensure consistent evaluation criteria. The results remain consistent with Table 3's findings.
>
> | Method | Real.  $\uparrow$   | Func.   $\uparrow$  | Lay.   $\uparrow$   | Comp.    $\uparrow$ |
> |-|-|-|-|-|
> | LayoutGPT| 5.83 | 6.21 | 5.26| 5.99|
> | I-Design| 6.6 | 6.57 | 5.73     | 6.79   |
> | Holodeck| 6.70| 7.35 | 6.67 | 7.45  |
> | Ours  | 8.80| 8.85 | 8.55| 8.98 |
>
> ### Q2: accessibility of the furniture
>
> This is a good suggestion.  We think it is a good point to add this aspect into a new physical metric. Here we refer to PhyScene and SceneEval and add additional physical metrics “accessibility”.
> Due to the time limits, here we randomly choose 3 bedrooms of each method and check the percentage of large furniture (such as bed, wardrobe, and nightstand) that can be reached in the front.
>
> | Method| Accessibility |
> |-|-|
> | LayoutGPT | 77% |
> | IDesign   | 80%|
> | Holodeck  | 82% |
> | ours  | 82%|
>
> Our methods shows high accessibility since the agent is able to check the walkable area. For example, the step 4 in Figure 1 shows the agent has recognized the crowding area and fix it. Note the Holodeck shows relatively high accessibility. That’s because its room has larger size than other methods and objects are closed to the wall.
>
> ### Q3: Release more details and limitations such as diversity of results and anti-physical content.
>
> **Diversity**
> The diversity of our method is high, due to the extendable tool cards and generation ability of foundation model. We implement the user study to evaluate the diversity of scenes.
>
> Specifically, we increase the generated scene number from 1 to 3 for each method in the second comparison setting in order to 1) reduce individual variance and 2) check diversity of different generation system.
> We ask the participants to choose:
>
> - Which method do you prefer?
> - Which method has greater diversity?
>
> | Method   |   w/ LayoutGPT | w/ I-Design | w/ Holodeck |
> |-|-|-|-|
> | Preference  | 94.30% | 91.40%  | 87.40%|
> | Diversity | 95.60% | 98.90%  | 90.00%|
>
> Results shows our methods has greater diversity while also stand for higher preference, surpassing other methods in a large degree.
>
> **More Limitations**
>
> - Asset retrieval may return poor results due to non-standard open-vocabulary formats and language embedding corner cases.
> - For assets not generated by Infinigen's procedures, supporting surfaces are estimated from geometry. If the geometry is irregular, this can result in tilted surfaces and unusual placement of child objects.
> - The previous tool produced sparse layouts inside containers, unable to densely arrange objects because it couldn't accurately detect supporter shapes. To address this, we extended the tool to enable denser placements, such as adding 30 well-arranged books on a shelf.
>
> ### Q4: More practical metrics：Shift in simulation
> Thanks for your suggestion.   To assess object stability insimulation, we measured:
> - Shift thresholds: percentage of objects moving >0.1m or >0.01m in 3s
> - Average displacement: Mean shift distance (meters)
>
> | Method |  > 0.1m  | > 0.01m | Avg Shift  |
> |-|-|-|--|
> | ATISS | 35.40% | 51.40%  | 0.356 |
> | DiffuScene  | 26.20%  | 39.30%  | 0.190  |
> | Physcene| 9.70%  | 19.60%  | 0.069  |
> | LayoutGP | 39.20% | 52.80%  | 0.477   |
> | IDesign | 5.00% | 11.50% | 0.041 |
> | Holodeck | 17.60% | 42.50%  | 0.113  |
> | Ours | 1.00%| 10.37% | 0.011 |
>
> The lowest shift of our mthods verify our great performance on the original physical metrics #OB and #CN. Obove all, our generated scene remains the most stable in simulation although with the largest object number.
>
> ### Q5：generation with more complex prompts for robot task
>
> This is a good point. The robotic task context can be achieved by adding specific description prompt design. Here we show some generated samples of complex prompts in Figure A3. Our method is capable of following complex requirements in different context.  It shows the controllability of objects’ category (basket, washtub, etc.), number (”A laundromat with 10 machines”), position (”a car in the center”), and precise components (decoration on the wall, small objects).
>
> ### Q6: Evaluate the impact of a single tool
>
> Table 5 shows that adding different types of tools (Initializer, Refiner, Implementor) greatly improves results. To assess the effect of a single tool, results can be compared with and without that tool using identical prompts, which requires multiple generations for reliable statistics. Due to time constraints, we do not provide precise quantitative results here. However, we observed that without the “Update Rotation” tool, the model is less sensitive to invalid rotations, reducing the “layout” score. Meanwhile, the “Add Crowd” tool lets supporters be densely filled with child objects (e.g., a shelf packed with books), which improves scene realism. If there is multiple tools in similar function, adding/removing a single tool will not make big difference. And if the tool has negative effect, the agent will recognize it during iterations and avoid using the tool.
>
> ### Q7: Room structure & single room scale
>
> We remove windows and doors to simplify the generation process, though our method can handle such structures and update the scene accordingly. Due to the format limitation of rebuttal, we can show some examples in the final version.
>
> For room scale, we focus on single rooms to improve scene quality as prompted by the user. The process can be repeated to handle multiple rooms. We could also generate multi-room scenes at once with some coordinate transformation.
>
> ### Q8: time limits
>
> The average time for a single iteration is 8.6 minutes, and the average time for generating a complete scene is 64 (min:35, max:130) minutes. Simpler and smaller room need less generation time. Here are some choices to reduce the time cost:
> - reduce the usage frequency of the physical optimization
> - remove some time-consuming tools, such as digital cousins. Although the results will be influenced, it will not cost too much drop.
> - reduce the iteration number.
>
> ### Q9: unrealistic scaling/placement for functional items
>
> That is a good point. The problems comes from the non-standard open-vocabulary dataset in two points. On one side, the object in open-vocabulary dataset such as Objaverse has no standard size, for example, a toy can be 10 meters in length. Thus resize is neccessary for those assets to eliminate the gap and fit the realness. On the other side, the front direction of the asset is not provided in the dataset, thus we utilize VLM to predict the front direction, which we find with high success rate, but it still failed in some cases since sometimes the front direction of the asset can not be defined uniformly, such as the elliptical bike, our expected front direction  is mistaken as the side direction, leading to unreasonable orientation and size.
> How to balance the standardization and generality for the open-vocabulary dataset is an unsolved problem, but it still provide a feasible and promising view for general scene synthesis.
>
> ### Q10: Articulations and squashing.
> That’s a good point. We will pay attention to the articulation and squashing in the future work.
>
> ### Q11: limited complexity of relationships
> The relationship is taken from the Infinigen with 3 types of obj-room relations and 7 types of obj-obj relations. This covers most of the common relationship. For more "well aligned" content, the code is also easily extendable for more complex and detailed relation.
>
> ### Q12: Implementation of the  tool & executor
> We list the implementation of each tool.
>
>  | Tool Name | Role | Use of Existing Works  | Our Contribution   |
> |--|-|--|--|
> | Init MetaScenes | Init scene with Real2Sim dataset  | MetaScenes| Choose data + convert format|
>  | Init PhyScene | Init scene with pretrained model  | PhyScene/DiffuScene/ATISS   | Choose data + convert format  |
>  | Init GPT | Init scene with LLM    | LLM  | Prompt engineeri   |
> | Add ACDC| Add tabletop objects visually| Stable Diffusion + ACDC| Significant changes on digital cousin|
>  | Add GPT | Add objects with LLM  | VLM | Prompt engineering|
> | Add Crowd  | Add crowded layout   | LLM+Infinigen| Utilize Infinigen rules + design module |
> | Refiner | Remove Object| Remove invalid objects| LLm   | Prompt engineering |
>  | Add Relation | Add relations to objects | VLM+ Infinigen  | Utilize Infinigen relations + prompt engineering |
>  | Update Rotation  | Fix rotation problems   | VLM | Prompt engineering|
> | Update Size | Rescale objects   | LLM  | Prompt engineering |
>  | Update Layout| Update improper layouts  | VLM | Prompt engineering |
>
> During the planning process, the agent is provided with brief description of each tool, shown in A3-A13 in supplementary material and use the function calling to choose a single tool for each step. Then it will run the tool itself and go through the executor to update the scene.
>
> Note we not only contribute on the tool itself, but also modify the infinigen to :
> - update iteratively
> - interactive in blender in realtime with socker
> - fit each tools rather than generate scene procedurelly.
> - Physical optimization
> - Add 3D marks and 2D top-down rendering
> - …
>
> ### Q13: prompt engineering
>
> We appreciate your insightful observation regarding the prompt engineering efforts in our work. We follow the structure of **Octotools** and design prompts for planner, tool cards,  executor, and reflector, which makes the framework  accurate, stable, and easily extensible. Other baseline methods (IDesign, LayoutVLM, Holodeck) also design several prompt for each step, which is a common choice for using the LLM and agent for generation task.
>
> ### Q14: Typos error
> Thanks for pointing out these problems. We will fix the typo error in the final version.

---

> > ### Comment · Reviewer_Er4e · 2025-08-06
> >
> > Thank you, authors, for your thoughtful response to my review and the additional details in answer to my questions.
> >
> > Overall I am satisfied with the extra information provided in the rebuttal and I hope that the expanded discussions and clarifications there-in will make it into the final manuscript. My rating will remain a 4 at this time.
> >
> > 1. Good to see that the expanded user study agrees with the initial smaller sample and I appreciate the effort of that expansion.
> >
> > 2. It is promising that the accessibility metric score is high for this approach. I agree that larger rooms and sparser arrangements will bias that metric, so good qualitative callout there. Also work noting that accessibility to the front of an object is not necessarily the definitive metric. You mount a bed from the side, for example. Rather, an object has interaction points which should be accessible. I think this feature will be much more relevant for embodied task use down the line and should be investigated further.
> >
> > 3. User study is ok for diversity metric in the short term, but it would be great to be able to quantify diversity better moving forward. How do we describe diversity of digital scenes in general? Perhaps developing some categorical metrics which can be evaluated automatically across axes: shape, size, layout connections, colors, materials, object classification distributions, etc...
> >
> > Thank you for expanding on limitations. I think with a system like this, your observations of the detailed limitations from building and running the system will be valuable to others looking to leverage or continue development of this approach. I hope you will consider adding these and any others you can think of the final manuscript.
> >
> > 4. Thanks for adding the shifting metric. I assume this is objects popping or dropping when physics is enabled. If so, I'm happy to see this approach coming out ahead. I think this is a critical feature for leveraging these scenes interactively.
> >
> > 5. I'd love to see some targeted examples of applying this system to generate a scene capable of supporting a provided robotic task. For example, if the task is to "take the dishes from the living room table to the kitchen sink" I would imagine the current solution could support that task. However, as the task becomes more specific or the language more complex, can a feasible scene creation be extracted? If so, we could see a new offline automated training paradigm.
> >
> > 6. I feel that much of the quality of the final product is a result of the tool mixture. While I generally agree with the qualitative explanations here, it would still be great to see which of the tools are producing the greatest metric shifts. I do understand the limitations of compute to fill that results matrix and runtime is a different issues altogether. Perhaps something to consider addressing in the future.
> >
> > 7. Windows/doors/multi-room scenes: I'm glad to hear this was considered and I do think it would be great to show some examples in the final version, even if those extensions were not the primary focus at this stage of the project.
> >
> > 8. Thank you for sharing concrete ballpark generation times, I think that information is critical for anyone considering applications based on this work and including more details like this improves transparency of the exposition.
> >
> > 9. I can understand this issue and the challenge with establishing standard coordinate systems for generic asset datasets. For now it would be good enough to be up front about these issues and mention them in limitations. In the future, I hope we as a community can solve this problem either in dataset annotation or a generic approach to address non-annotated asset registration.
> >
> > 10. Future work acknowledged, would be good to mention this.
> >
> > 11. Future work acknowledged, would be good to mention this.
> >
> > 12. Thanks for the expanded details. Hopefully this will be included in the final manuscript.
> >
> > 13. Acknowledged.
> >
> > 14. You're welcome, happy to help improve the final exposition.

---

### Official Review · Reviewer_kZsN · 2025-07-03

**Clarity:** 3
**Significance:** 3
**Originality:** 2
**Rating:** 5
**Confidence:** 4

**Summary:**

This submission introduces SCENEWEAVER, a reflective agentic framework that aims to unify scene synthesis paradigms through tool-based, iterative refinement. The framework employs an LLM-based planner to iteratively build a 3D scene via a suite of extensible scene generation tools. It performs self-evaluation of physical plausibility, visual realism, and semantic alignment with user input. SCENEWEAVER brings the reason-act-reflect paradigm to 3D scene synthesis and the authors claim that it is the first agentic framework for this purpose. They perform experiments on open-vocabulary scene synthesis to demonstrate its effectiveness compared to existing methods with respect to visual realism, physical plausibility, and instruction following. They also provide ablation studies to provide mechanistic insight into the proposed framework.

**Questions:**

As alluded to in the weaknesses section, I believe the manuscript would improve with some attention to the evaluation. Is there a justification for why a higher average number of objects in the scene (#Obj) is better? Could #Obj be used to weight #OB/#CN (as it stands to reason there are more opportunities for errors given a higher object count)?
I like the inclusion of the human study, especially the pair-wise comparison. It would be powerful to add some statistical weight to this aspect of the evaluation by increasing the sample size and demonstrating additional statistics like inter-annotator agreement.

**Ethical Concerns:**

["NO or VERY MINOR ethics concerns only"]

**Final Justification:**

The authors have addressed my concerns and their responses to the other reviewers has been instructive. The proposed revisions to the work regarding limitations and the improved user study confirm my original view and understanding of the work. As a result, I maintain my rating.

**Limitations:**

Yes

**Paper Formatting Concerns:**

Typo on line 31 "lac" should be "lack"

**Quality:**

3

**Strengths And Weaknesses:**

The strength of this submission lies in the integration of the synthesis tools (both generation-focused and evaluation-focused). Because these tools are seamlessly integrated in the framework, the ReAct-style reasoning and planning pipeline for 3D scenes becomes tractable. And because LLMs are now driving the synthesis, open-vocabulary becomes a strength of the generation process. While self-reflection (or at least consistency checks) in 3D is not wholly novel, connecting these established paradigms in a meaningful way through tool use is compelling. That being said, without the ability to review the code, the extensibility of the framework remains unclear.
The two weaknesses of this submission both lie in the evaluation. First, leaning on VLM-as-a-judge for visual and semantic evaluation seems problematic as the same model is being used as both agent and judge (this is setting aside that VLMs are known to struggle with spatial reasoning). While it has become common practice, I would like to see stronger citations than 2 preprints as a basis for adopting it.   Second, it is unclear to me that the chosen metrics for physical evaluation are desirable (specifically the number of objects in a scene). Furthermore, claiming best performance when there is a tie (as seems to be the case often with out-of-boundary objects (#OB), and collided object pairs (#CN)) seems incorrect.

---

> ### Author Rebuttal · Authors · 2025-07-30
>
> Thank you for the valuable feedback and suggestions.
> ### Q1: Explanation of physical metrics
> We refer to PhyScene to evaluate out-of-boundary objects `#OB` and collided object pairs `#CN`, which indicate the reasonbale layout from two aspects. Although there is a tie between these 2 metrics, we could also achieve the best performance for both of them, indicating the effectiveness of a well-designed agent framework. Meanwhile, we refer to IDesign and evaluate the number of objects `#Obj`. A higher number of objects means more objects to interact for embodied AI and also leads to higher diversity of the generated scene.
>
> ### Q2: Increase sample size and inter-annotator agreement in user study.
> **User Number and Sample Size**
>
> In the main paper, we following prior work LayoutVLM for the user study, thus we only recruited **5** students. To make the user study more solid, we improve the user number to **20** and increase the evaluated scene number to **200**, each of them is shown  40 samples.  We show them the same instructions used as prompt in the LLM reflector for the participant to evaluate the scene carefully.
> We statistic the results into the following table, which indicates similar results with the previously shown Table 3 in the main paper.
>
> | Method      | Real.  $\uparrow$   | Func.   $\uparrow$  | Lay.   $\uparrow$   | Comp.    $\uparrow$ |
> |-------------|-----------|-----------|-----------|-----------|
> | LayoutGPT   | 5.83     | 6.21    | 5.26    | 5.99    |
> | I-Design    | 6.65     | 6.57     | 5.73     | 6.79     |
> | Holodeck    | 6.70     | 7.35     | 6.67     | 7.45     |
> | ours        | 8.80    | 8.85     | 8.55     | 8.98     |
>
> **Score Alignment and Validity of Evaluation**
>
> Note the evaluation results of human-evaluation and LLM-score can not be exactly the same, thus a slight difference between them is a normal phenomenom.
> To further confirm the stability of these justification method, we check the ailgnment between different users as well as LLM.  We convert the users' and LLM's scores on different scene to rankings and calculate Kendall’s Tau for the 4 metrics:
>
> | Alignment        | Real.   $\uparrow$  | Func. $\uparrow$    | Lay.  $\uparrow$    | Comp. $\uparrow$    |
> |------------------|---------|-------|-------|-------|
> | User-User     | 0.43	| 0.42	|0.45	|0.40|
> | User-LLM        | 0.46	| 0.45	|0.48|	0.55|
>
> Results show strong user-user agreement and user-LLM agreement on different metrics. And we find that user-LLM has higher alignment score than user-user, which indicates that LLM is more stable than user in evaluation. These alignment results also demonstrate the effectiveness of using LLM for justification in Table 2/3 of the main paper.
>
> **Diversity**
> The diversity of our method is high, due to the extendable tool cards and generation ability of foundation model. We implement the user study to evaluate the diversity of scenes.
>
> Specifically, we increase the generated scene number from 1 to 3 for each method in the second comparison setting in order to 1) reduce individual variance and 2) check diversity of different generation system.
> We ask the participants to choose:
>
> - Which method do you prefer?
> - Which method has greater diversity?
>
> | Method   |   w/ LayoutGPT | w/ I-Design | w/ Holodeck |
> |--|--|--|--|
> | Preference  | 94.30% | 91.40%  | 87.40%  |
> | Diversity | 95.60% | 98.90%  | 90.00%     |
>
> Results shows our methods has greater diversity while also stand for higher preference, surpassing other methods in a large degree.
>
> ### Q3: stronger citations for VLM judgement
> Thanks for this suggestion.  We have checked the preprinted citations in our submission, and the **LayoutVLM** and **FirePlace** has been accepted to CVPR 2025, which utilize VLMs for 3D spatial reasoning. Additionally, **GPTEval3D**[1](accepted by CVPR 2024) proves that VLM can serve as a Human-Aligned Evaluator for Text-to-3D Generation. **SceneCraft**[2] (accepted by ICML 2024) also utilizes VLM to critic and revise the scene with Blender script. We will update the citation in the final version.
>
>
>
> [1] Tong Wu, Guandao Yang, Zhibing Li, Kai Zhang, Ziwei Liu, Leonidas Guibas, Dahua Lin, and Gordon Wetzstein. Gpt-4v(ision) is a human-aligned evaluator for text-to-3d generation. In CVPR, 2024.
>
> [2] Hu Z, Iscen A, Jain A, et al. Scenecraft: An llm agent for synthesizing 3d scenes as blender code. ICML, 2024.

---

> > ### Comment · Reviewer_kZsN · 2025-08-04
> >
> > I would like to thank the authors for their responses as well as their work to improve and solidify the user study.
> >
> > Q1: Explanation of physical metrics
> > While I appreciate the efforts to tie physical, quantitative metrics to the generation and the references behind them, I believe my point still stands. Explaining the rationale behind the metrics would be beneficial, even just including the sentence: “A higher number of objects means more objects to interact for embodied AI and also leads to higher diversity of the generated scene.” Would be sufficient.
> > I still hold that a ratio or weighting could be helpful to balance these quantitative metrics against one another: If #Obj is higher, there are more opportunities for collisions or out-of-boundary objects right?
> > To that end, when I see Holodeck in Table 2, for instance, score [32.2, 0, 0] on [#OBJ, #OB, #CN], I view that as superior to [14, 0, 0] for SceneWeaver. That is why I suggest it is misleading to claim best performance in that table. As an aside, it does then highlight the strength of the proposed method for open-vocabulary detection, which is expected.
> >
> > Q2: Improved user study
> > I’d like to thank the authors for the additional effort in assembling this user study. Not only does it underscore the strengths of SceneWeaver but it is a very valuable touchpoint for the community in a world where automated/computer judging is becoming the norm.
> >
> > Q3: VLM-as-a-Judge citations
> > Thank you for finding and updating these references. Pairing these references with the user study is a nice way of holistically evaluating the course we are walking down as a community.

---

> > > ### Author Response · Authors · 2025-08-06
> > >
> > > Thank you for the response.
> > >
> > > Here we check the `#CN` of bedroom generated by Holodeck again and find that we have previously made a mistake in the data. We apologize for this mistake. And the correct result should be [32.2,0,**38.5**], which means the collided pairs are 38.5 in average for bedroom.
> > >
> > > According to this, we can find that the number of collision pairs `#CN` increase rapidly with the number of objects in Holodeck. The `#CN`/`#Obj` ratio for both bedroom (0.22) and living room (0.97) of Holodeck are very high, while ours keeps zero.
> > >
> > > | Method   | #Obj (Bedroom) | #CN (Bedroom) | #CN/#Obj (Bedroom) | #Obj (Living Room) | #CN (Living Room) | #CN/#Obj (Living Room) |
> > > |----------|---------------|---------------|--------------------|--------------------|-------------------|------------------------|
> > > | Holodeck | **32.2**          | 38.5          | 0.97               | **23**                 | 5.3               | 0.22                   |
> > > | Ours     | 14            | **0**             | **0**                  | 17.3               | **0**                | **0**                      |
> > >
> > > We will correct the mistake and update the result in the final version.

---

### Note · Authors · 2025-08-13

Dear Area Chair and Reviewers,


Thank you for the thorough review. We thank all the reviewers for their feedback and helpful suggestions.

During the rebuttal session, we have added additional user studies and several experiments to confirm the **outstanding performance and diversity** of our method. New physical metrics (e.g., shift in simulation, accessibility) were conducted to further demonstrate the **physical plausibility**. We also claim that our method has **extensive capabilities**, such as generating multi-room scenes and delicate room structures like windows and doors. More qualitative results, limitations, and details on tool implementation will be supplemented in the final version, as discussed.

To address remaining concerns:

**Q1. Claims in Table 1 (DJFe, dp6k):**

We appreciate the reminder to clarify criteria.

* Real: Defined as realistic layouts considering human daily habits.
* Large-scale: Methods able to generate infinite scene variations.
* Accurate: Methods with well-defined rules ensuring explicit object relations and strict placement.

For example, Infinigen's realism relies on simple rule, texture and rendering, but layout issues remain: (1) monitors may face walls, as placement rules lack direction awareness; (2) random sampling without spatial priors can scatter desks. Therefore, we do not consider Infinigen “real.” We will revise Table 1 and improve justifications.

**Q2. LLM judgment (DJFe):**

Reviewer DJFe expressed concerns about GPT’s judgments in both the agent’s iterations and the final validation. We understand the concern and could add it into the limitations, but we regard this as an acceptabe solution in our work. **On one hand**, GPT-based evaluation is more cost-effective and stable (as discussed) than human studies and provides a practical choice for evaluation. **On the other hand**, using GPT’s judgments during iteration also demonstrates the unique ability our reflective agentic framework. Other baseline methods with fixed pipelines and limited tools lack the ability to adjust results based on evaluation scores or LLM feedback, while ours makes it possible to iteratively refine scenes through feedback-driven planning with modular tools.

Above all, we thank the reviewers for their support of our work. We hope these clarifications address the remaining questions and facilitate the decision.

---

### Decision · Program_Chairs · 2025-09-17

**Decision:**

Accept (poster)

**Comment:**

The paper presents an LLM-driven method for creating synthetic 3D scenes. The reviewers appreciate the proposed pipeline which closes the loop in ensuring physical plausibility and includes self-evaluation (during synthesis). While some concerns were raised during the discussion period, these were mostly addressed. The main outstanding issues is the use of the LLM as a final evaluation metric for reporting quantitative performance. It was noted that while there is recent precedence in the literature for LLM-based evaluation, the approach is fundamentally flawed especially where the same evaluation metrics are used during refinement of the generation process. The Area Chair agrees with this assessment. The end result being that the metrics merely test the correctness of the iterative process rather than the overall (independent) performance of the model itself. The authors should clearly identify this shortcoming in the paper. There are also minor concerns around naming of performance metrics that should be clarified in subsequent revisions of the paper. Overall, however, the contributions of the work were appreciated and warrant publication.